

# The Climate Impact of Hypersonic Transport

Johannes Pletzer[1,2], Didier Hauglustaine[3], Yann Cohen[3], Patrick Jöckel[1], and Volker Grewe[1,2]

[1]Deutsches Zentrum für Luft- und Raumfahrt, Institut für Physik der Atmosphäre, Oberpfaffenhofen, Germany
[2]Delft University of Technology, Kluyverweg 1, 2629 HS Delft, The Netherlands
[3]Laboratoire des Sciences du Climat et de l'Environnement, LSCE-IPSL (CEA-CNRS-UVSQ), Université Paris-Saclay,
91191 Gif-sur-Yvette, France

**Correspondence:** Johannes Pletzer (johannes.pletzer@dlr.de)

**Abstract.** Hypersonic aircraft flying at Mach 5 to 8 are a means for travelling very long distances in extremely short times and even significantly faster than supersonic transport (Mach 1.5 to 2.5). Fueled with liquid hydrogen (LH2) their emissions consist of water vapour ($H_2O$), nitrogen oxides ($NO_x$) and unburnt hydrogen. If LH2 is produced in a climate- and carbon neutral manner, carbon dioxide does not have to be included when calculating the climate footprint. While $H_2O$ that is emitted

near the surface has a very short residence time (hours) and thereby no considerable climate impact, super- and hypersonic aviation emit at very high altitudes (15 km to 35 km), with residence times of months to several years, and therefore the emitted $H_2O$ has a substantial impact on climate via high altitude $H_2O$ changes. Since the (photo-)chemical lifetime of $H_2O$ is largely decreasing at altitudes above 30 km via the reaction with $O(^1D)$ and via photolysis, one could speculate that $H_2O$ climate impact from hypersonics flying above 30 km becomes smaller with higher cruise altitude. Here we use two state-of-the-art

chemistry-climate models and a climate response model to investigate atmospheric changes and respective climate impacts due to two potential hypersonic fleets flying at 26 km and 35 km, respectively. We show for the first time that the (photo-)chemical $H_2O$ depletion at high altitudes is overcompensated by a recombination of hydroxyl radicals to $H_2O$ and an enhanced methane depletion, leading to an increase in $H_2O$ concentrations. This results in a steady increase of the $H_2O$ perturbation lifetime of up to $4.41 \pm 0.20$ years at 35 km. We find a $0.083 \pm 0.014$ % and $0.16 \pm 0.015$ % depletion of the ozone layer and a $43.0 \pm 4.8$ Tg

and $94.0 \pm 4.5$ Tg increase in stratospheric $H_2O$ due to the two hypersonic fleets flying at 26 km and 35 km respectively. Our calculations show that the climate impact of hypersonic transport is estimated to be roughly 8-20 times larger than a subsonic reference aircraft with the same transport volume (revenue passenger kilometers) and that the main contribution stems from $H_2O$.

## 1  Introduction

The climate impact of aircraft emissions has been studied for decades and becomes more and more important. Estimated aviation growth rates will increase aviation's contribution to climate change and by that challenge the support of the Paris Climate Agreement (UN, 2015; Terrenoire et al., 2019; Grewe et al., 2021, Planes et al., 2021). A recent study estimates the contribution of aircraft activity to human-made climate change to 3.5 (3.4, 4.0) %, which is an estimate based on effective



radiative forcing and aviation fuel use (2011) (Lee et al., 2021). Furthermore, while carbon dioxide ($CO_2$) effects contribute a third to the effective radiative forcing, non-$CO_2$ effects contribute two thirds. This estimate is based on current aircraft fleets that are powered with kerosene and fly at altitudes from 10 to 12 km. With these aircraft, it takes travellers approximately one day to fly around the world.

Two development goals for future aircraft fleets are on one hand to reduce climate impact and on the other hand to reduce
travel time. As an example, roadmaps for a more climate-friendly liquid-hydrogen based aviation industry have been developed in agreement with existing research and put the potential reduction in climate impact of 50-75 % (Fuel Cells and Hydrogen: Joint Undertaking, 2020). But this mainly addresses the concept of subsonic aircraft flying at 10-12 km altitude. Higher altitudes are especially interesting for high-speed aircraft concepts that promise customers to save a considerable amount of travel time, especially on middle- and long-range flights. Aircraft designs in that category are super- and hypersonic aircraft. These aircraft
are designed to travel at higher speed and higher altitudes compared to conventional subsonic aircraft and could reduce travel time around the globe to some hours. In turn, however, flight altitude and fuel type, which depend on aircraft design, can have a strong influence on atmospheric composition and thus on climate impact. The climate impact of high-speed aircraft designs has been addressed for supersonic aircraft flying in the lower stratosphere with kerosene as well as with cryogenic fuels (IPCC, 1999; Gauss et al., 2003; Grewe et al., 2007, 2010).

Impact of supersonic aircraft at stratospheric altitudes is often discussed in terms of ozone concentration changes and ultra-violet radiation and extensive overviews on these topics have recently been published (Zhang et al.; Tuck, 2021; Matthes et al., 2022). Additionally, there are multiple publications on the quantitative climate impact of supersonic aircraft fleets. A review of a selection of research programs including a direct comparison of radiative forcings (RF) was given in 2010 by Grewe et al., who estimated the ratio of RF from supersonic to subsonic aircraft to be 3 (S4TA fleet, eight passengers), 6 (Airbus fleet,
250 passengers) and 14 (Boeing fleet, 309 passengers). The latter two numbers were adapted by the author from Grewe et al. (2007) and IPCC (1999), respectively. While these numbers initially appear to differ greatly, the authors present a correlation of flight altitude (range from 15-20 km) and RF of non-$CO_2$ effects and additionally state that supersonic climate impact can approximately be scaled with fuel consumption. Hence, cruise altitude (i.e. speed which in turn influences fuel consumption) clearly is a crucial factor for climate impact.

At even higher altitudes, in the middle to upper stratosphere, hypersonic aircraft travel at MACH 4 or more and the emitted trace gases are reacting in another atmospheric environment compared to sub- and supersonic aircraft. Depending on cruise altitude and latitude of potential routes, hypersonic aircraft could fly in the upper parts of or above the ozone layer. As reference, the ozone ($O_3$) mixing ratio is largest at around 31 km at tropical regions and has no clear peak at polar regions (Tegtmeier et al., 2013). Notably, the climate impact of hypersonic aircraft emissions is largely depending on their atmospheric residence time.
More specifically, the residence time of species emitted by aircraft in the stratosphere is mainly controlled by the large-scale circulation of air (Brewer-Dobson Circulation), their chemical interaction with stratospheric air constituents and photolysis. All three processes are highly dependent on altitude. The amount of emitted species is determined by aircraft and engine design, fuel type and trajectory layout. Kerosene, as the conventional fuel, is a technical option for hypersonic aircraft, but the initial focus in hypersonic studies is often on cryogenic aircraft. In general, cryogenic aircraft can be powered by pure gases



such as methane ($CH_4$) or hydrogen ($H_2$) that are cooled to the liquid phase with the goal of reducing volume, i.e. increasing range per tank fill, and for other technical advantages (Peschka, 1998). Hence, compared to kerosene, cryogenic fuel could in theory be particle-free and would not have an indirect aerosol effect (Ponater et al., 2006). One of the few potential cryogenic fuels is liquid natural gas (LNG), which consists mostly of $CH_4$. Similar to kerosene-fueled aircraft, LNG ultimately increases atmospheric $CO_2$ concentrations. In comparison to $H_2$ fuel, LNG's direct climate impact could be slightly lower at specific

altitudes (Grewe et al., 2017, cruise altitude approximately 13 km). However, the $CO_2$ perturbation originating from fossil fuel is subject to a large variety of sinks with different lifetimes. In general, the range is approximated with 2-20 centuries, where most of the $CO_2$ climate impact is taken up by ocean and biosphere sinks and 20-35 % remain in the atmosphere for longer time (Archer et al., 2009). On the other hand, $H_2$ fueled aircraft emit mainly water vapor ($H_2O$), whereas nitrogen oxides ($NO_x$) and $H_2$ are byproducts, with the latter depending on combustion efficiency. Their lifetimes are hours to years for $H_2O$ and years

for $NO_x$ and $H_2$ (Brasseur and Solomon, 1984; Johnston et al., 1989; Grewe and Stenke, 2008; Ehhalt and Rohrer, 2009), which is substantially lower than for $CO_2$. Hence, a lifetime at least an order of magnitude shorter is a reason why currently $H_2$ fuel is often the preferred climate-friendly option for hypersonic aircraft designs. Another reason is the potentially more efficient (photo-)chemical destruction of $H_2O$ at high altitudes, reducing the residence time and thus climate impact of emitted $H_2O$. More specifically, several studies suggest that chemical reaction of $H_2O$ with $O(^1D)$ and photolysis could efficiently

remove emitted $H_2O$ at upper stratospheric altitudes and higher (Brasseur and Solomon, 1984; Steelant et al., 2015). This could potentially create a synergy of aircraft flying at these altitudes and $H_2$ propulsion aimed at reducing climate impacts. Thus, hypersonic aircraft with $H_2$ propulsion are seen as a potentially more climate friendly alternative to supersonic aircraft. For completeness, it is important to mention that the choice of fuel type does not only include climate impact, but is a trade-off between, for example, energy content, i.e. best range of aircraft, or cooling properties for thermal regulation (Blanvillain and

Gallic, 2015).

  However, the quantitative climate impact of hypersonic aircraft, regardless of fuel type, has not been assessed with a global atmospheric model yet and how it compares to supersonic aircraft still remains to be answered as well. Especially $H_2$ fueled aircraft are important to look at due to the broad discussions about global $H_2$ infrastructure and in consideration of the fact that $H_2O$ is a potent greenhouse gas that affects ozone at stratospheric altitudes (Stenke and Grewe, 2005). Recent publications on

impact by hypersonic aircraft are few and focus on $H_2$ fueled aircraft. Ingenito (2018) published an estimate based on a fleet of 200 aircraft flying at 30 km of type LAPCAT II MR2.4 (Steelant and Langener, 2014, Long-Term Advanced Propulsion Concepts and Technologies), quantifying the reduction of atmospheric ozone with $3.6 \ 10^{-4}$ % and a temperature increase due to $H_2O$ to 142 mK in one year. Another study by Kinnison et al. (2020) was done with a global atmospheric model to assess the impact of hypersonic fleet emissions on the ozone layer at 30 and 40 km altitude. They focus on sensitivity of stratospheric

ozone to local perturbations of $NO_x$ and $H_2O$. Their estimate on a reduction of atmospheric ozone with the same amount of fuel is 4.0 % and 2.2 % at 30 km and 40 km, respectively.

  In this study we evaluate the climate impact of non-$CO_2$ effects from hypersonic aircraft and its altitude dependency with the metric RF. In particular, we present the impact of $H_2O$, $NO_x$ and $H_2$ emissions by hypersonic aircraft driven with $H_2$ fuel on atmospheric composition. The results are based on the comparison of simulations with two chemistry climate models.



The hypersonic emission data this study is using, was developed in the 'HIgh speed Key technologies for future Air transport
- Research & Innovation cooperation scheme'-project (HIKARI, Blanvillain and Gallic (2015)), whose authors recommend
$H_2$ fuel as a first choice. The economic and technical requirements of two fleets of hypersonic aircraft considered there will
provide the estimate of the altitude dependency of climate impact of hypersonic aircraft emissions. Two advanced hypersonic
fleet scenarios (Technology Readiness Level 1-3), differing in cruise altitude, cruise speed and thus fuel consumption, will

allow the comparison of climate impacts of technically viable hypersonic aircraft concepts. The two hypersonic aircraft designs
considered here are the models ZEHST (Zero Emission High-Speed Transport) and LAPCAT PREPHA-type (Falempin et al.,
1998; Scherrer et al., 2016, French National Research and Technology Program for Advanced Hypersonic Propulsion) with
cruise altitude at approximately 26 km and 35 km, respectively. The paper is structured as follows. After the introduction, we
present the two chemistry-climate models EMAC (ECHAM/MESSy; European Centre HAMburg general circulation model;

Modular Earth Submodel System) and LMDZ-INCA (Laboratoire de Météorologie Dynamique; INteraction with Chemistry
and Aerosols) and their setups. In section 3 we present the HIKARI emission inventory and the temporal evolution of aircraft
emissions. After a model evaluation with aircraft measurements in section 4, we show the atmospheric composition changes in
section 5. The climate impact is part of section 6 'Radiation and Climate'. In the end, a discussion and summary is included.



## 2 Methods and Simulations

We have performed simulations of atmospheric changes caused by emissions from hypersonic transport with the two Atmospheric Chemistry General Circulation Models (AC-GCMs) EMAC and LMDZ-INCA in order to address model dependencies of the results. Differences and conformity between model setups that were used in this study are presented in Table 1. It lists the key properties of both models.

EMAC consists of the spectral dynamical core of the GCM ECHAM5 by the Max Planck-Institute for Meteorology (Roeck-
ner et al., 2006, version five) and MESSy, able to combine all components relevant for Earth System Models, published by Jöckel et al. (2005, 2010, 2016).

The LMDZ-INCA global chemistry-aerosol-climate model couples on-line the LMDZ General Circulation Model (version six, Hourdin et al., 2020) and the INCA model (version five, Hauglustaine et al., 2004; 2014). The interaction between the atmosphere and the land surface is ensured through the coupling of LMDZ with the ORCHIDEE (ORganizing Carbon and
Hydrology In Dynamic Ecosystems, version 1.9) dynamical vegetation model (Krinner et al., 2005).

More information on the model set-ups are presented in the following sub-chapters.

**Table 1.** Table stating the key properties of model setups for EMAC and LMDZ-INCA as applied for this study.

| Model | EMAC[1] | LMDZ-INCA |
|---|---|---|
| GCM | ECHAM | LMDZ |
| Vertical limits [km] | 0-80 | 0-80 |
| Vertical limits [hPa] | Surface-0.01 | Surface-0.04 |
| Nudging limits [hPa] | Surface-10.0 | Surface-1.0 |
| Nudging data | ERA-Interim | ERA-Interim |
| Nudging relax. time | 6-48 hrs | 2.5 hrs |
| Grid cells (lon,lat,lev) | 128 x 64 x 90[2] | 144 x 143 x 39 |

[1]ECHAM5 v5.3.02, MESSy v2.54.0, Jöckel et al. (2010). [2]Named T42L90MA
where T42 refers to a triangular truncation at wave number 42, corresponding
to a quadratic Gaussian grid of approximately 2.8 ° by 2.8 ° in latitude and
longitude, L90 refers to 90 vertical hybrid pressure levels and MA to 'Middle
Atmosphere'.

While the model setups of EMAC and LMDZ-INCA are consistent with respect to the model domain and nudging data there are differences with respect to the vertical resolution, the nudging relaxation time and the model domain where nudging is applied.

Both models were intensively validated, e.g. the dynamics at stratospheric altitudes by the model's stratospheric age of air and by tropical upward mass flux. Further details on the respective models are given in the following subsections.



## 2.1 EMAC Model Set-Up

The EMAC model setup is based on that of simulation RC1SD-base-10, recommended due to it's affirmative agreement with observations, especially ozone, and described in detail by Jöckel et al. (2016). The model results were compared extensively with ERA-Interim. Deviations from model to observations are a cold bias with a vertical maximum at 200 hPa and values of $\pm$ 4 K (Fig. 12, Jöckel et al. (2016)). A comparison with satellite data shows that over the annual cycle ozone volume mixing ratios are well reproduced in the stratosphere, apart from southern polar regions and with larger differences at tropospheric altitudes. An IAGOS-CARIBIC (In-service Aircraft for a Global Observing System; Civil Aircraft for the Regular Investigation of the atmosphere Based on an Instrument Container: Petzold et al., 2015) data comparison for the upper troposphere–lower stratosphere (UTLS) shows deviations around 5 % and larger values up to 30 % in June-September for regions above the tropopause and an overestimate of up to 40 % in the troposphere. Methane, carbon monoxide and acetone concentrations are underestimated by the model, specifically at tropospheric altitudes. Transport by the Brewer-Dobson-Circulation was tested by comparing the model tropical upward mass flux to ERA-Interim and mean age of stratospheric air (AoA) to MIPAS observations. While the largest differences for AoA are at polar latitudes and the upper stratosphere, the model still shows overall a reasonable agreement, especially for the lower stratosphere and extratropics, and the tropical upward mass flux shows the best agreement for setups with specified dynamics (nudging) and without global mean temperature nudging (Jöckel et al., 2016). Summarizing, while there are deviations, the setup including nudging is among the others especially well suited for stratospheric sensitivity studies.

A common setup with small signal-to-noise ratio for sensitivity studies is the Quasi-Chemistry-Transport-Model (QCTM) mode published by Deckert et al. (2011). As this study assesses the impact of emitted $H_2O$ (as well as $NO_x$ and $H_2$), a decoupling of atmospheric dynamic, chemistry and radiation was technically not possible when adding $H_2O$ to the model's hydrological cycle. However, we achieved a sufficiently large signal-to-noise ratio by Newtonian relaxation (nudging) towards ERA-Interim reanalysis data for the years 2000-2014. To summarize the nudging setup briefly, nudging is applied to the prognostic variables divergence, vorticity and logarithm of the surface pressure, but not to global mean temperature and has that in common with RC1SD-base-10 from Jöckel et al. (2016). As mentioned before the latter allows the cold bias to appear in the simulation. The respective relaxation times are listed in 1. Further information on the very similar nudging process, apart from different relaxation times, is described in the next subsection about LMDZ-INCA.

Prognostic variables were accounted for by the MESSy submodel TENDENCY to tag submodel contributions like e.g. cloud-, advection- or emission-processes to specific humidity (Eichinger and Jöckel, 2014). Additionally, $H_2O$ emissions were added to the model's hydrological cycle with a new MESSy submodel (H2OEMIS), which uses either TENDENCY (in this study) or directly adds water vapour perturbations to the specific humidity tracer. An overview of all active submodels can be found in the appendix.



### 2.1.1 ECHAM5/MESSy Submodel H2OEMIS

$H_2O$ and the associated hydrological cycle play an important role in atmospheric radiation and dynamics and thereby the
general circulation. As it is a precursor of the atmospheric hydroxyl radical (OH), it also largely controls atmospheric chemistry.
For earlier versions of the model, the chemistry calculations operated with an $H_2O$-tracer that had to be kept synchronous with
the prognostic specific humidity of the underlying GCM. In recent versions, the chemistry feedback on the hydrological cycle
now directly alters the specific humidity. Due to this it was not possible to include offline $H_2O$ for altering the prognostic
specific humidity directly. But the modular structure easily allowed the development of the new submodel H2OEMIS to include
the possibility to directly emit $H_2O$ into the atmosphere and alter specific humidity. Briefly summarized, gridded $H_2O$ emission
flux data is imported into the model via the IMPORT submodel (Kerkweg and Jöckel, 2015), then H2OEMIS converts the flux
into a tendency of the specific humidity and applies this tendency to the prognostic variable qm1, i.e. specific humidity in
MESSy. The submodel and further information are available with the MESSy release version 2.55.0. Two short movies can be
found in the video supplements showing emitted $H_2O$ by hypersonic aircraft and it's addition to the specific humidity (zonal
mean representation and world map view respectively).

### 2.1.2 Additional Diagnosis of Chemical Destruction and Production of $H_2O$ with MECCA

The net amount of chemical production and destruction of $H_2O$ is significant for the concentration of $H_2O$ in the stratosphere.
Above 20 km the largest contribution comes from transport through the tropical tropopause layer via the deep branch of the
Brewer-Dobson-Circulation and oxidation of $CH_4$, which was recently reconfirmed with EMAC model studies by Eichinger
et al. (2015a, b); Frank et al. (2018) and satellite studies by Noël et al. (2018). In EMAC the chemical mechanism is applied
among others via the MESSy submodel MECCA (Module Efficiently Calculating the Chemistry of the Atmosphere) developed
by Sander et al. (2005, 2011). This specific submodel is able to include tracers that keep track of production as well as
destruction of chemical reactants. Thus we included five tracers for chemical destruction of $H_2O$ and 45 tracers for chemical
production of $H_2O$ for a deeper understanding of underlying processes. Further information on the $H_2O$-specific reaction
rates and all other reaction rates are given in the supplement of this study (file *meccanism.pdf, scavinism.pdf*) and beyond
that in the supplement of Jöckel et al. (2016). While MECCA considers gas- and heterogeneous-phase reactions in the tropo-
and stratosphere, the submodel SCAV (SCAVenging) includes aqueous phase reactions in clouds and precipitation and the
corresponding induced removal of trace gases and aerosols by wet deposition (Tost et al., 2006), CH4 (Winterstein and Jöckel,
2021) issues $CH_4$ oxidation (while MECCA feedbacks to specific humidity), MSBM (Multiphase Stratospheric Box Model)
calculates the polar stratospheric cloud-chemistry (Jöckel et al., 2010), DDEP (Dry DEPosition) and SEDI (SEDImentation)
are responsible for dry deposition and sedimentation of aerosols, respectively (Kerkweg et al., 2006a), AIRSEA addresses air
and ocean surface interaction (Pozzer et al., 2006), OFFEMIS (OFFline EMISsion) and ONEMIS (ONline EMISsion) add
prescribed and online-calculated emissions (Kerkweg et al., 2006b) and LNOX (Lightning Nitrogen OXides) includes $NO_x$
production by lightning, where we used the 'Grewe' coupling parameterisation as described in Tost et al. (2007); Grewe et al.
(2001). The resulting total lightning $NO_x$ for the baseline simulation is 0.2 TgN/yr.





## 2.2  LMDZ-INCA Model Set-Up

In the present LMDZ-INCA configuration, we use the "Standard Physics" parameterization of the GCM (Boucher et al., 2020).
The model includes 39 hybrid vertical levels extending from the surface up to 80 km. The horizontal resolution is 1.9 degrees
in latitude and 3.75 degrees in longitude. The primitive equations in the GCM are solved with a 3 min time-step, large-scale
transport of tracers is carried out every 15 min, and physical and chemical processes are calculated at a 30 min time interval.
For a more detailed description and an extended evaluation of the GCM we refer to Hourdin et al. (2020). The large-scale
advection of tracers is calculated based on a monotonic finite-volume second-order scheme (Hourdin and Armengaud 1999).
Deep convection is parameterized according to the scheme of Emanuel (1991). The turbulent mixing in the planetary boundary
layer is based on a local second-order closure formalism. The transport and mixing of tracers in the LMDZ GCM have been
investigated and evaluated against observations for both inert tracers and radioactive tracers (e.g., Hourdin and Issartel, 2000;
Hauglustaine et al., 2004) and in the framework of inverse modelling studies (e.g., Bousquet et al., 2010; Zhao et al., 2019).

INCA initially included a state-of-the-art CH4-NO$_x$-CO-NMHC-O$_3$ tropospheric photochemistry (Hauglustaine et al., 2004;
Folberth et al., 2006). The tropospheric photochemistry and aerosols scheme used in this model version is described through
a total of 123 tracers including 22 tracers to represent aerosols. The model includes 234 homogeneous chemical reactions, 43
photolytic reactions and 30 heterogeneous reactions. Please refer to Hauglustaine et al. (2004) and Folberth et al. (2006) for
the list of reactions included in the tropospheric chemistry scheme. The gas-phase version of the model has been extensively
compared to observations in the lower-troposphere and in the upper-troposphere. For aerosols, the INCA model simulates the
distribution of aerosols with anthropogenic sources such as sulfates, nitrates, black carbon (BC), organic carbon (OC), as well
as natural aerosols such as sea-salt and dust. The aerosol component of the LMDZ-INCA model has been extensively evaluated
during the various phases of AEROCOM (e.g., Gliss et al., 2021; Bian et al., 2017).

Earlier versions of the LMDZ-INCA model including gas phase tropospheric chemistry only have been previously used to
assess the impact of subsonic aircraft on tropospheric ozone (Koffi et al., 2010; Hauglustaine and Koffi, 2012). This version of
the model has been extended to include an interactive chemistry in the stratosphere and mesosphere (Terrenoire et al., 2022).
Chemical species and reactions specific to the middle atmosphere have been included in the model. A total of 31 species
were added to the standard chemical scheme, mostly belonging to the chlorine and bromine chemistry, and 66 gas phase
reactions and 26 photolytic reactions. Water vapor is now affected by both physical and chemical processes in LMDZ. In the
stratosphere, an additional tracer is introduced in order to account for photochemical production and destruction in INCA. In
addition, heterogeneous processes on Polar Stratospheric Clouds (PSCs) and stratospheric aerosols are parameterized in INCA
following the scheme implemented by Lefevre et al. (1994). The excess of H$_2$O and HNO$_3$ is removed from the gas phase
when saturation occurs and is used to compute the surface area concentration in the PSC region. Heterogeneous reaction rates
are calculated explicitly, as a function of the surface area available, mean molecular velocity, and the reaction probabilities.
The model does distinguish between types I and type II PSCs. Furthermore, the PSC scheme includes sedimentation of the
cloud material. The sedimentation of PSC particles affects the vertical distribution H$_2$O, HNO$_3$, and HCl. Condensed species
are returned to the gas phase when clouds evaporate. In the presence of PSCs, the heterogeneous reactions convert bromine and



chlorine reservoirs (HCl, HBr, ClONO$_2$, BrONO$_2$) into reactive species (Cl$_2$, ClNO$_2$, HOCl, Br$_2$, BrNO$_2$, HOBr) based on 9 additional heterogeneous reactions introduced in the chemical scheme. The distribution of stratospheric aerosols is prescribed according to the CCMI exercise (Chemistry-Climate Model Initiative, Thomason et al., 2018).

In this study, meteorological data from the European Center for Medium-Range Weather Forecasts (ECMWF) ERA-Interim reanalysis have been used to nudge the GCM winds. The relaxation of the GCM winds towards ECMWF meteorology is
performed by applying at each time step a correction term to the GCM $u$ and $v$ wind components with a relaxation time of 2.5 hours (Hourdin and Issartel, 2000; Hauglustaine et al., 2004). The ECMWF fields are provided every 6 hours and interpolated onto the LMDZ grid.

The anthropogenic emissions from the Shared Socioeconomic Pathways scenarios (SSPs) (scenario SSP3-7.0) prepared by Gidden et al. (2019) are used and added to the natural fluxes used in the INCA model. The $ORCHIDEE$ vegetation model
has been used to calculate off-line the biogenic surface fluxes of isoprene, terpenes, acetone and methanol as well as NO soil emissions as described by Lathière et al. (2006). The lightning NOx emissions are parameterized in the model based on convective cloud heights as described in Jourdain et al. (2001). Based on this parameterization, the total lightning NO$_x$ emissions for the baseline simulation is 5.5 TgN/yr.

## 2.3 Simulations

Both models simulate a time period of more than fourteen years. This includes a ten to twelve year spin-up phase to achieve a chemical and dynamic equilibrium that takes into account long lifetimes at stratospheric altitudes. Lower boundary conditions, direct- and traffic-emissions of simulations are based on IPCC's RCP6.0 scenario (CCMI) for EMAC and the SSP3-7.0 scenario for LMDZ-INCA. This includes the surface mixing ratios for methane, nitrous oxide as well as chlorine and bromine containing species and excludes air traffic, where we used an emission inventory from the HIKARI project. The RCP6.0 and SSP3-7.0
scenarios are very similar and the latter can be treated as an updated version (IPCC CMIP6) of the former (IPCC CMIP5). The beginning of air fleet operation including hypersonic aircraft was set to 2050. Therefore, the assumptions of the RCP scenario for other emissions are the fifteen year period 2050-2064. The assumptions of SSP are fixed on their 2050 values. In summary, on one hand, the simulations were carried out with nudged dynamics allowing a reduction of the signal-to-noise ratio from the time period 2000-2014, and, on the other hand, the atmosphere's chemical composition of simulations is based on assumptions
from 2050 (and onwards) i.e. when hypersonic aircraft technology is potentially ready for commercial use.




## 3 The HIKARI Emission Inventory and Spin Up

### 3.1 Emission

HIKARI was an international project of Europe and Japan for high-speed transport resulting in a potential timeline for further development of high-speed transportation up to commercial operation (Hikari means 'Light' in japanese, 2015). Blanvillain

and Gallic published the roadmap study in 2015, which combines economic viability, environmental constraints as well as technological requirements. We use the formerly unpublished trade-off emission scenarios by HIKARI. These include two hypersonic scenarios with aircraft ZEHST and LAPCAT.

- ZEHST is short for Zero Emission High-Speed Transport and is a high-speed aircraft project, which includes a strategy to reduce environmental impact with a zero $CO_2$ emission policy. The aircraft is based on a ramjet engine for cruise

phase and travelling speed is at Mach 4-5. This aircraft is developed for 60 passengers and an intermediate transport range of approximately 9 000 km (e.g. Paris-Tokyo, Defoort et al. (2012)).

- LAPCAT is short for Long-Term Advanced Propulsion Concepts and Technologies. It is a joint effort of many European institutes, resulting among others in an aircraft model meant to travel at Mach 8 with scramjet technology, carry up to 300 passengers and for long-range flights of approximately 18 000 km (e.g. Brussels-Sydney, Steelant and Langener

(2014); Steelant et al. (2015)). The LAPCAT version is based on the technology level developed in the French high speed propulsion program $PREPHA$.

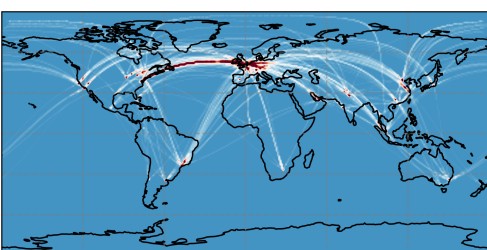

**Figure 1.** World map of $H_2O$ emission location (white to red) by LAPCAT aircraft summed over all vertical levels. A colorbar is not included because the motivation is to depict specifically the emission location. Only one plot representative for all HIKARI emission scenarios is shown, because hypersonic features of emission are dominant in the vertical sum and ZEHST and LAPCAT scenarios show a quite similar horizontal distribution on the world map. A vertical distribution is shown in Fig. 2.



While we focus on $H_2$ driven aircraft in this study, the HIKARI emission inventory includes carbon-based emission data as well. In general, $H_2$ driven aircraft with air-breathing engines emit $H_2O$ and $NO_x$, but potentially includes a rest of unburnt $H_2$ fuel. For the HIKARI emission inventory 10 % of unburnt $H_2$ is assumed.

In total, the HIKARI emission inventory contains three scenarios. In addition to the subsonic reference scenario based on an Airbus A350, a mixed-fleet scenario with ZEHST-type hypersonic aircraft and a mixed-fleet scenario with LAPCAT-type hypersonic aircraft are part of this inventory (Fig. 2). The respective cruise altitudes are approximately 12, 26 and 35 km.

The collection of annual emissions of trace gases for the three scenarios are listed in Table 2. This includes the global emissions as well as the proportion emitted at stratospheric altitudes and the latter mostly consists of trace gases emitted at
the cruise phase of hypersonic aircraft. Approximately two-thirds of $H_2O$ are emitted at stratospheric altitudes for both, the ZEHST and LAPCAT scenario, and one-third in the troposphere. The main part of $NO_x$ is emitted at tropospheric altitudes and $H_2$ is emitted to 76 % and more at stratospheric altitudes. Another property shown in the same table, market penetration, is a measure of how many of the global flight routes are suited for the specific hypersonic aircraft compared to subsonic aircraft. Hence, there is a smaller market for the aircraft LAPCAT whose design is able to travel extremely large distances
(approximately 18,000 km) due to the limited selection of appropriate city pairs like Brussels-Sydney. In comparison, ZEHST is a potential faster alternative for 25 % of the subsonic aviation market with a smaller range (approximately 9,000 km). The aircraft designs differ in passenger seats with 60 for ZEHST and 300 for LAPCAT. In Fig. 2 (b) and (c) emission features at altitudes 6-11 km are visible. This originates from acceleration and climb to cruise altitude by hypersonic aircraft of ZEHST and LAPCAT.

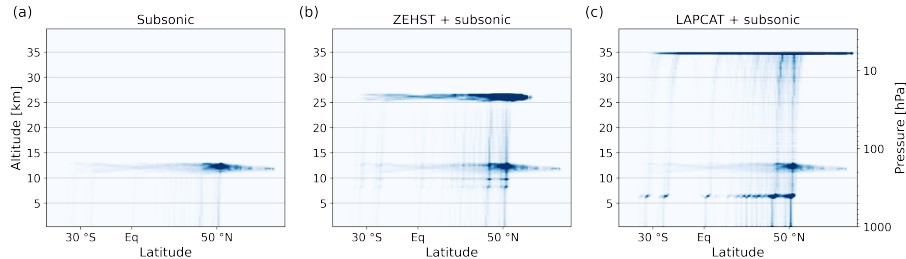

**Figure 2.** Zonal sum of trace gas emission location. (a) Depicts the subsonic reference scenario, (b) depicts the ZEHST scenario with a hypersonic market penetration of 26.0 % and (c) depicts the LAPCAT scenario with a hypersonic market penetration of 9.8 %. A colorbar with units is not shown, since the location (altitude, latitude) is the interesting piece of information here. For annual amounts of emissions see Table 2. The data is taken from the HIKARI project's emission inventory.





**Table 2.** HIKARI emission inventory. Annual emission of trace gas species $H_2O$, $NO_x$ and $H_2$. The upper three rows contain amounts of trace gases emitted in the whole atmosphere, while the lower three rows contain amounts of trace gases emitted at stratospheric altitudes only (above 100 hPa). Note that for ZEHST and LAPCAT a part of the subsonic aviation is replaced by hypersonic transport and hence the numbers depict whole air traffic scenarios.

| Scenario | Domain | $H_2O$ [Tg yr$^{-1}$] | $NO_x$ [TgNO$_2$ yr$^{-1}$] | $H_2$ [Tg yr$^{-1}$] | Market penetration[3] |
|---|---|---|---|---|---|
| Subsonic | global | 5.022 | 0.072 | 0.0 | 0 % |
| ZEHST | global | 21.581 | 0.113 | 0.163 | 26.0 % |
| LAPCAT | global | 31.366 | 0.115 | 0.307 | 9.8 % |
| Subsonic | above 18 km | 0.0 | 0.0 | 0.0 | - |
| ZEHST | above 18 km | 13.741 | 0.020 | 0.153 | - |
| LAPCAT | above 18 km | 21.237 | 0.031 | 0.236 | - |

[4]Market penetration depicts the ratio of subsonic to hypersonic aviation market, where hypersonic aircraft take over some of the subsonic market.

## 3.2 Spin Up and Temporal Evolution

The respective full aircraft fleet is in operation for the total simulated time of fifteen years. Annual emissions accumulate over the years and perturbation of emitted trace gas concentration eventually reach equilibrium at the multi-annual mean after 8-10 years. The emissions are balanced by transport and loss to tropospheric altitudes as well as (photo-)chemical losses and production. Five years of simulation in equilibrium remain for average values and significance tests. The monthly-mean mass perturbation above the tropopause (WMO, 1957) is shown in Fig. 3 over the simulation timeline from 2000 to 2015. EMAC and LMDZ-INCA show similar patterns of monthly oscillations. The oscillations of the latter are smoother, which potentially is related to the lower vertical resolution or the feedback between atmospheric composition and radiation in EMAC.

After fifteen years of continuous operation of hypersonic fleets, the total emission of trace gases for ZEHST and LAPCAT scenarios at stratospheric altitudes are 206 Tg and 319 Tg of $H_2O$, 0.3 Tg and 0.5 TgNO$_2$ of $NO_x$ and 2.3 Tg and 3.5 Tg of $H_2$, respectively. Most of the emitted trace gases are transported to tropospheric altitudes and only parts of the annual perturbation remain. More information on the equilibrium perturbation is presented in Section 5.





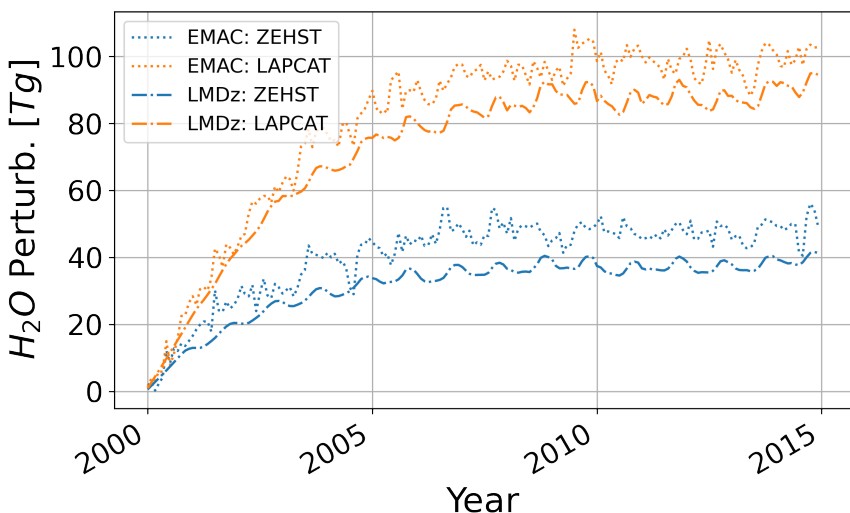

**Figure 3.** Temporal evolution of mass perturbation of $H_2O$ in Tg above the tropopause. Blue and orange lines represent scenario ZEHST and LAPCAT, respectively. The annual amount of emitted trace gases is listed in Table 2.

## 4  Model Evaluation

In this chapter, we evaluate the model results with observations from commercial aircraft. It is important to verify the models performance in the upper troposphere–lower stratosphere region, especially in the northern extratropics, where most of the

trace gases are emitted and the downward stratosphere-to-troposphere transport (SST) occurs. The SST, including the trace gases emitted at stratospheric altitudes, is a very important step of the continuous process, that eventually removes these trace gases from the atmosphere. A fact that is further emphasized by the results of this publication. IAGOS offers data particularly fit for this region of interest. In a forthcoming publication, we extend the evaluation to higher altitudes and compare EMAC model results to satellite data (Pletzer and Grewe, 2022, in prep). In addition, a comparison between LMDZ-INCA and ozone

soundings measurements has also been presented in Terrenoire et al. (2022).

### 4.1  Observation data set

The research infrastructure IAGOS provides in situ measurements on board a fleet of commercial aircraft. Observations of ozone and water vapour started in August 1994 and are still being collected so far. IAGOS mostly samples the UTLS in the northern extratropics, with cruise data spreading between 9 and 12 km above sea level. The ozone instruments are based on UV-

absorption spectrometry and their accuracy, precision and time response are 2 ppb, 2 % and 4 s, respectively. $H_2O$ is measured using a capacitive hygrometer. The latter's precision and time response are generally 5 %, or 6 % in the thermal tropopause at midlatitudes (Smit et al., 2014), for relative humidity and 5–300 s for $H_2O$, respectively, with regard to ice (Helten et al., 1998; Neis et al., 2015).



In order to allow a direct comparison between simulation outputs and the IAGOS data, the Interpol-IAGOS software used here (Cohen et al., 2021a) first projects observations onto model grids, then derives monthly means. The subsequent products are called IAGOS-DM-INCA and IAGOS-DM-EMAC. The -DM suffix refers to the distribution onto the model grid. Since the output from the models have a daily resolution, a mask is applied with respect to the IAGOS sampling. The subsequent products are called INCA-M and EMAC-M. The -M suffix refers to the mask. In this way, the monthly means derived from both, the IAGOS and the simulations data sets, represent the same days for each grid cell.

Seasonal and annual climatologies are then calculated on the 3D models grids. As in Cohen et al. (2021a), a grid point is filtered out, if the total amount of IAGOS data is below a minimum threshold, the latter decreasing with latitudes in order to account for the grid cell area. The validated grid points are then averaged together as partial columns, with a 400 hPa lower bound in order to exclude the IAGOS data recorded during ascent and descent phases near airports. In order to ensure a correct vertical representation, we select only the columns derived from at least two grid cells. Our assessment focuses on the northern midlatitudes since, first, the UTLS is far more influenced by the stratosphere in the extratropics than in the tropics and, second, only the UT is sampled in the tropics. The seasons defined in the northern midlatitudes are thus typically extratropical.

The scores used for the evaluation are the modified normalized mean bias (MNMB) and the Pearson correlation coefficient. For a set of N grid cells with an observed value $o_i$ and a simulated value $m_i$, the MNMB is defined as follows:

$$MNMB = \frac{2}{N} \sum_{i=1}^{N} \frac{m_i - o_i}{m_i + o_i} \qquad (1)$$

In contrast to the classical mean bias, that is sensitive to larger values, the MNMB treats large and small values with a similar sensitivity. Thus it is very valuable for an assessment in the UTLS without separating tropospheric and stratospheric air masses. Indeed, since the tropopause altitude varies geographically, the aircraft fleet will record an important geographical variability in both, ozone and water vapour.

## 4.2 Model comparison to IAGOS observations

The evaluation of the simulations from EMAC and LMDZ-INCA against IAGOS in the northern extratropics are synthesized in the Taylor diagrams shown in Fig. 4, and in Figs. A5–A8 for the seasonal scale. They are derived from the mean climatologies shown in Figs. A9–A13. Both model products are well correlated with the observations, with r ∼ 0.90 for water vapour in INCA-M, and r ∼ 0.95 in the other cases. Independent of the magnitude, the models capture the geographical variations well for ozone, water vapour and temperature. On annual average, the INCA-M product is characterized by a relatively weak mean bias in ozone, water vapour and temperature. The EMAC-M product has a systematic cold bias in the extra-tropics, with half of the grid cells ranging between -3.8 and -2.5 K. It leads to an upward shift of the tropopause and thereby underestimates ozone and overestimates water vapour volume mixing ratios. These two features are visible in Fig. A9, where it is shown to be more representative of the higher latitudes. Last, according to Figs. A5–A8, the ozone and water vapour biases keep the same sign through the seasons for EMAC-M, contrary to INCA-M. The ozone (respectively water vapour) MNMB are





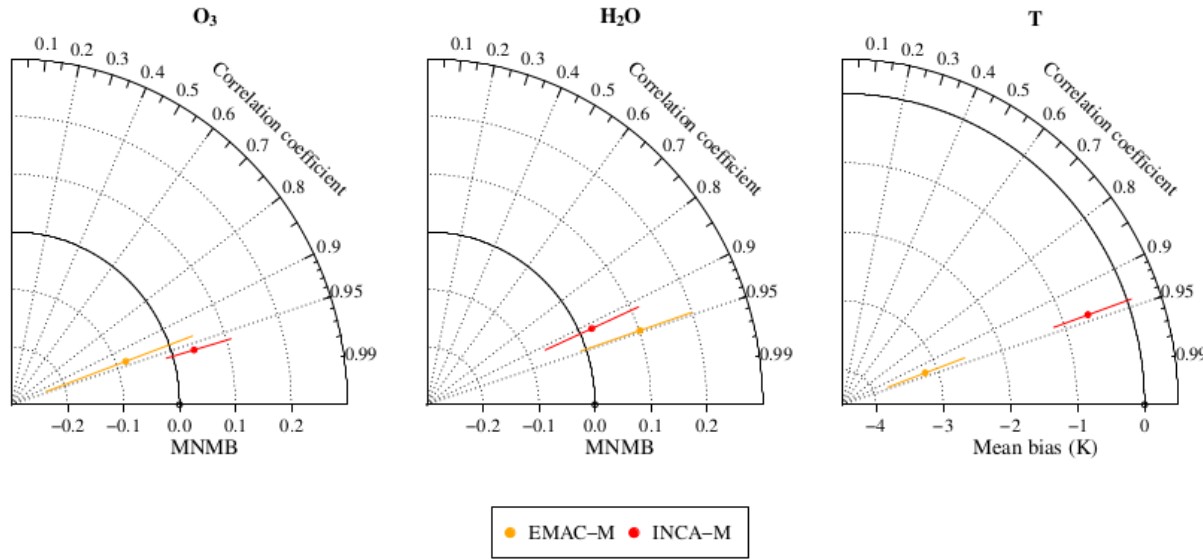

**Figure 4.** Taylor diagrams showing the assessment of the simulations against IAGOS in the northern extratropical UTLS. From left to right, ozone, water vapour and temperature are represented. For ozone and water vapour, the radial axis shows the modified normalized mean bias, and the mean bias for the temperature. The error bars represent the interquartile interval. The orthoradial axis displays the Pearson correlation coefficient.

particularly negative (respectively positive) in both models during summer, possibly suggesting an increased underestimation of the stratospheric influence on the UTLS during this season.

## 5   Atmospheric Composition Changes

### 5.1   Water Vapour

Stratospheric water vapour (SWV) stems mostly from upward transport at tropical latitudes and oxidation of $CH_4$. The main
factors for loss of SWV are reaction with $O(^1D)$, photolysis and the transport into the troposphere at the subtropical tropopause breaks. Note that those processes are resolved in our model simulations. Figure 5 shows the volume mixing ratio in equilibrium for the respective model and aircraft as a 5-year average (2010-2014). The SWV perturbation is clearly visible in both models, especially at the northern hemisphere with the maximum located at around 50-60° N, which overlaps with the maximum trace gas emission location. Overall, the perturbation patterns agree well between the models, especially for altitudes from 16 to 37
km (approximately 100-4 hPa) and with differences at higher altitudes 37 to 79 km (approximately 4-0.01 hPa). The latter has a negligible impact on the mass perturbation in the models, since the largest mass perturbation is located in the middle and lower stratosphere, where air density is larger. A t-test shows that all zonal-mean $H_2O$ perturbations are statistically significant



at a 99.9 % level and only parts in the tropical UTLS (EMAC: ZEHST, LMDZ-INCA: ZEHST, LAPCAT) do not reach that value.

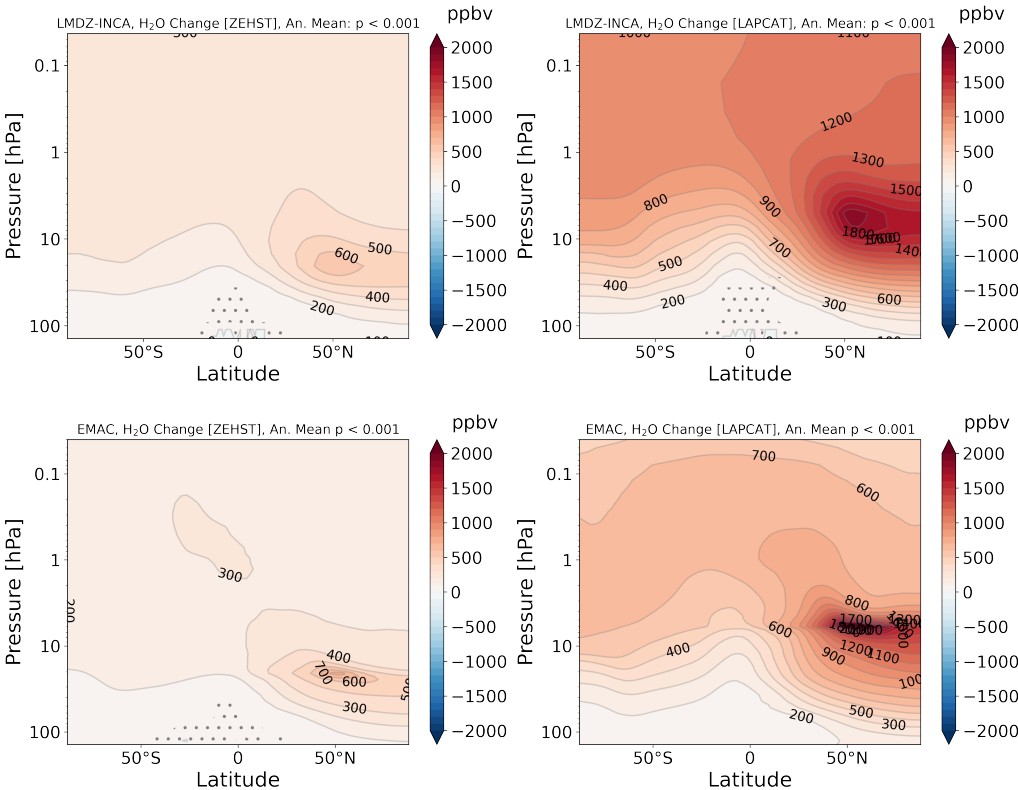

**Figure 5.** Multi-annual mean (2010-2014) of $H_2O$ perturbations (ppbv) for the ZEHST scenario (left) and the LAPCAT scenario (right) for both models LMDZ-INCA (upper) and EMAC (lower) after approximately thirteen years of continuous emission. Cruise altitudes are approximately at the respective perturbation maxima. The dotted region corresponds to the grid points where the mean perturbations are not significant at a 99.9 % level.

Absolute values of the mass perturbation and the respective lifetime of water vapour are listed in Table 3. Values were calculated for the perturbation above the tropopause (WMO, 1957). The $H_2O$ mass perturbation is approximately twice as large for the higher flying aircraft compared to the lower flying aircraft for each model and the perturbation lifetime clearly increases with altitude from 2.8-3.5 years to 4.2-4.6 years. Due to the difference in annual $H_2O$ emission of both aircraft, the perturbation lifetime scales differently compared to the mass perturbation.

Mass perturbation and perturbation lifetime are affected by the (photo-)chemical removal of emitted water vapour. The key processes are photolysis and reaction with $O(^1D)$. The combined average lifetime is shown in Fig. 6 for LMDZ-INCA (a) and EMAC (b). Photolysis clearly increases with altitude resulting in shortest lifetime at the upper end of the simulated altitude range and at the equator region, where incident sunlight is strongest on average. The reaction with $O(^1D)$ has a maximum at



**Table 3.** Perturbation and perturbation lifetime of $H_2O$ in teragram and years, respectively, for the ZEHST and the LAPCAT scenario and for each of the two models.

| Model/Scenario | Perturb. ZEHST | Perturb. Lifetime ZEHST | Perturb. LAPCAT | Perturb. Lifetime LAPCAT |
|---|---|---|---|---|
| LMDZ-INCA | 38.27 Tg | 2.79 yr | 89.45 Tg | 4.21 yr |
| EMAC | 47.82 Tg | 3.50 yr | 98.47 Tg | 4.61 yr |

around 45-50 km, where the loss of $O_3$ due to photolysis and thus concentrations of excited atomic oxygen $O(^1D)$ is generally
larger. The combination shows mainly the features of $H_2O + O(^1D)$, with contributions of photolysis at altitudes above 50 km.

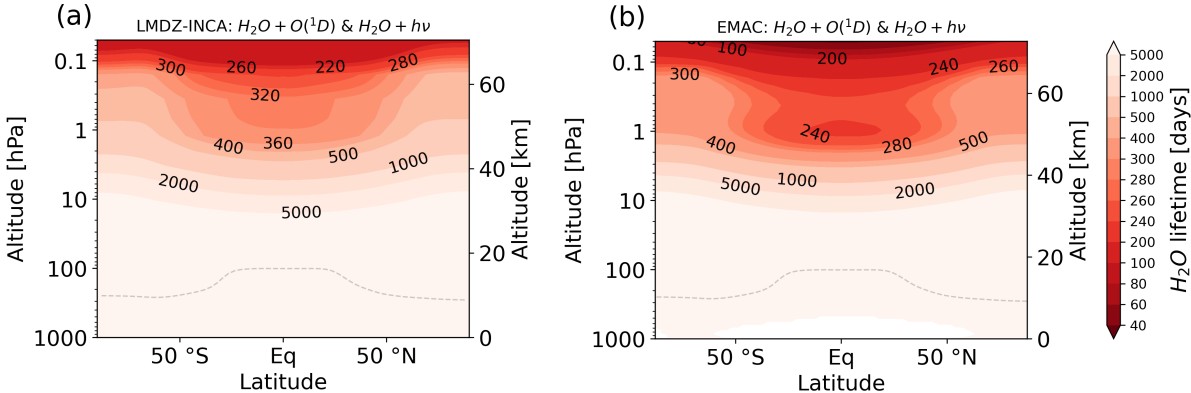

**Figure 6.** Zonal mean (photo-)chemical $H_2O$ lifetime in days dependent on photolysis and reaction with $O(^1D)$ for LMDZ-INCA (a) and EMAC (b). Figure A4, showing both reactions independently, can be found in the appendix.

Clearly, the (photo-)chemical destruction increases with altitude. However, our results show that the perturbation doesn't decrease with altitude. To verify the chemical loss and production of water vapour we introduced an additional diagnosis within EMAC. It is a budget calculation for all chemical loss and production terms with respect to water vapour. Further information on our setup can be found in the supplementary files *meccanism.pdf* and *scavinism.pdf* and the general explanation in the
supplement of Sander et al. (2005). We find that 29 % and 60 % of the annual emitted water vapour (Table 2) is destroyed for ZEHST and LAPCAT, respectively, which is generally in agreement with Fig. 6 that indicates larger (photo-)chemical $H_2O$ losses at higher altitudes. The absolute values of $H_2O$ loss are shown as dark blue bars in Fig. 7. The main drivers are photolysis and the reaction with $O(^1D)$ (upper blue bars), where the reaction with $O(^1D)$ dominates for both aircraft scenarios. The other reactants responsible for $H_2O$ destruction, $N_2O_5$, $ClNO_3$ and $BrNO_3$, are not significantly contributing. However, there is not
only loss but also production of water vapour, which even overcompensates the loss by 0.76 Tg and 1.95 Tg in the case of ZEHST and LAPCAT, respectively (dark red bars). This equals an increase of the initial annual perturbation above 18 km by



5.53 % for the former and 9.32 % for the latter. The resulting change from emissions and (photo-)chemistry is balanced by transport to the troposphere.

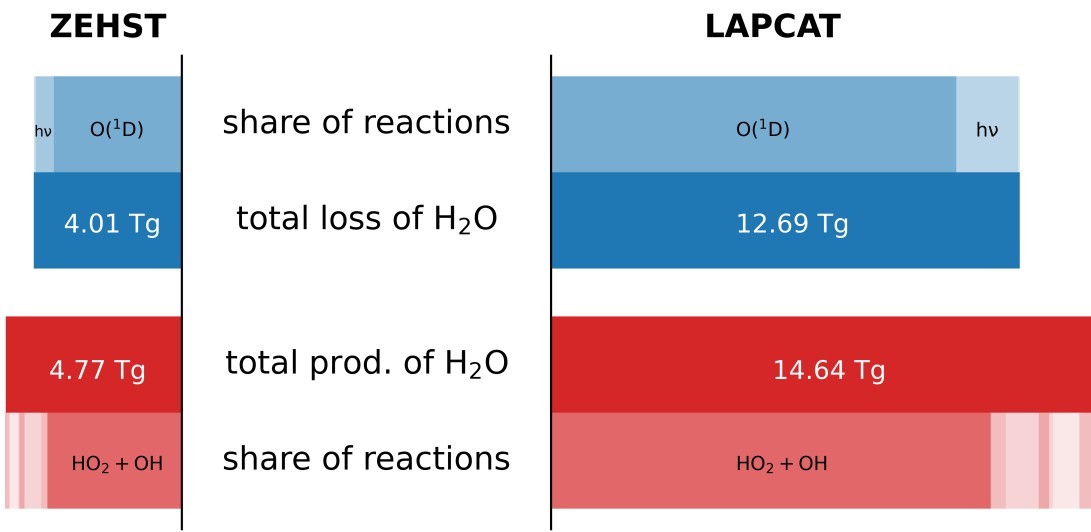

**Figure 7.** Bar plot of annual chemical loss and production perturbation of $H_2O$ at stratospheric altitudes (100-0.1 hPa) for the ZEHST (left) and the LAPCAT (right) scenarios due to emitted trace gases. This diagnosis includes a total of five $H_2O$ destroying and 45 $H_2O$ producing chemical reactions. The most relevant reactions of loss and production are shown as bright red and blue bars. Additionally, all dominant reactions of production are presented in detail in Fig. 8.

We call the surplus of production over destruction net-recombination. This net-recombination originates from different

sources and the significant reactions for production are shown in Fig. 8. Absolute values of production, 4.77 Tg/yr and 14.6 Tg/yr for ZEHST and LAPCAT, originally from Fig. 7 are shown here with additional information. In total, 45 reactions are contributing to $H_2O$ production and we grouped the reactions in four different categories, which are reactions with C-, N-, Cl- and H-O-compounds. The main contributors by far is the $HO_x$-cycle (green) followed by either more efficient methane oxidation for LAPCAT (red) or contributions of nitric acids $HNO_3$ and $HNO_4$ for ZEHST (blue).

The ratio of the categories is different for the two altitudes. H-O-compounds contribute more for emission at the higher altitude. Opposite to the expected removal of $H_2O$ emissions, we found a before unknown net-recombination of $H_2O$. Both models show an increase in $H_2O$ perturbation lifetime and $H_2O$ perturbation at the higher altitude and this analysis indicates that a net-recombination and enhanced methane depletion is overcompensating the $H_2O$ destruction. Our finding is robust with good agreement between the two models.





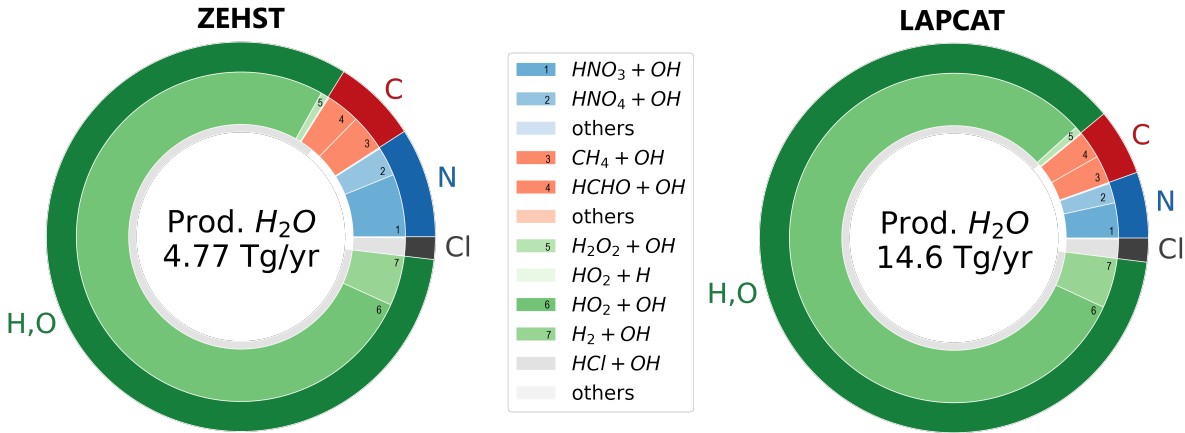

**Figure 8.** Pie plots on chemical reactions responsible for $H_2O$ production derived from the EMAC simulation results. Left: ZEHST scenario. Right: LAPCAT scenario. The darker outer ring shows the molecular category the reactions belong to (carbon-, nitrogen-, chlorine- or hydrogen-based). The total sum of $H_2O$ production is written in the center. The colored inner ring shows the proportion of different reactions within each category. We added black numbers in the legend and the bright inner ring to represent the relation in addition to colours.

## 5.2 Nitrogen Oxides $NO_x$ and Ozone $O_3$

Continuous emission of $NO_x$ ($NO + NO_2$) of hypersonic aircraft has a significant impact on ozone chemistry. The family of perturbed NO-compounds is collectively described as $NO_y$ ($NO_x$ + and their nitrogen reservoir species). While $NO_x$ is very reactive in catalytic cycles of ozone chemistry, $NO_y$ additionally includes more stable molecules, like nitric acid ($HNO_3$), that act as a sink and remove NO-compounds from catalytic ozone cycles for a longer time compared to $NO_x$. Figure 9 shows the perturbation of $NO_x$ for each model and each aircraft fleet. In general, the perturbation patterns are similar, but EMAC results show a more detailed perturbation pattern. For LMDZ-INCA we see a general $NO_x$ increase, whereas in EMAC, additionally, a decrease is visible at approximately 1 hPa upwards ranging from equatorial regions to midlatitude regions for LAPCAT. In comparison, LMDZ-INCA shows one cluster of $NO_x$ perturbation originating from the emission location (purple bar), while EMAC shows several (some not significant) clusters. The clusters locations seem to overlap, but the larger emission of LAPCAT makes it difficult to distinguish the clusters, as it covers the clusters interspace. The clusters appear at two levels, i.e. at cruise altitudes and at higher altitudes, just below 1 hPa. Results for the lower flying aircraft are more often outside of the 5 % uncertainty margin in comparison to the higher flying aircraft. This may be related to the larger emission of the latter, resulting in a larger perturbation which differs from zero perturbation with higher confidence.

As mentioned before, $NO_x$ are very reactive in catalytic cycles of ozone chemistry. The perturbation of ozone resulting mainly from $H_2O$ and $NO_2$ emission is shown in Fig. 10. The correlation between the increase of $NO_x$ and the decrease of $O_3$ is clearly visible in the patterns at mid-stratospheric altitudes and especially at the cruise altitudes. LMDZ-INCA shows a slight



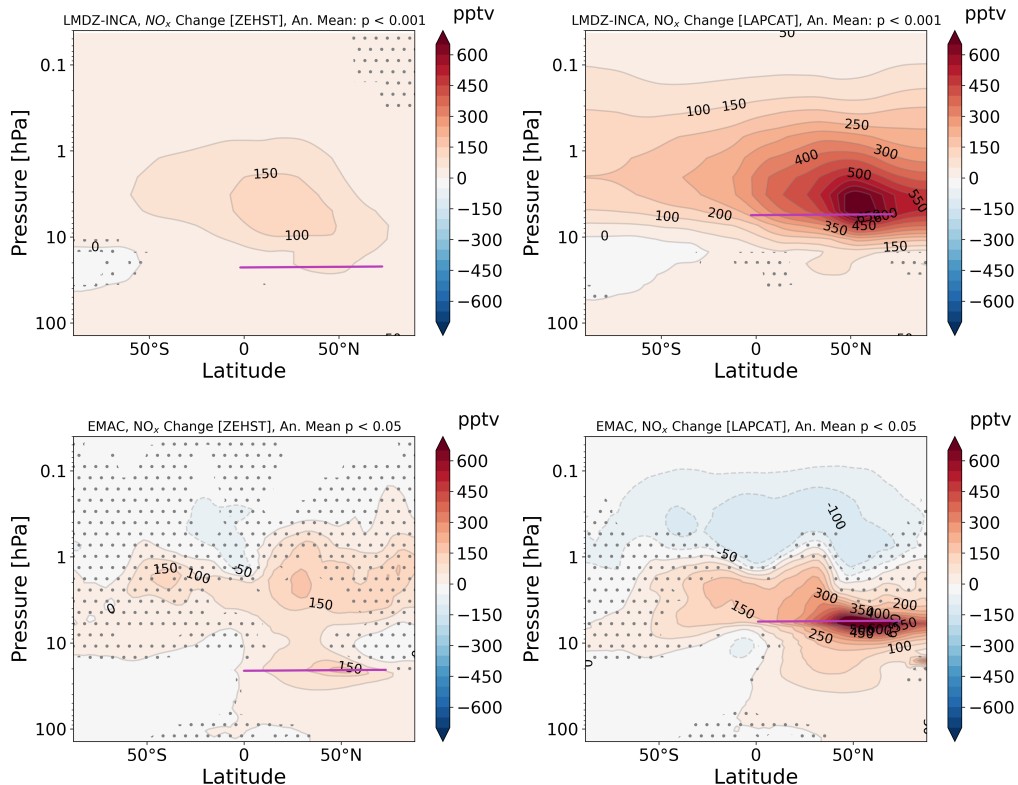

**Figure 9.** Annual mean (2010-2014) of $NO_x$ perturbation ($\mathrm{pptv}$) for the ZEHST scenario (left) and the LAPCAT scenario (right) for both models LMDZ-INCA (top) and EMAC (bottom). Purple lines represent respective cruise altitudes. Hatched areas are characterized by a p-value higher than the threshold indicated in the title of each plot.

increase originating from the tropical UTLS and a decrease with multiple clusters everywhere else, apart from no perturbation at lower latitudes at the highest altitude. EMAC results show a higher vertical resolution pattern with clusters of $O_3$ increase and decrease, which may be due to the higher vertical resolution of EMAC (90 vertical grid levels) compared to LMDZ-INCA

415     (39 vertical grid levels) in this study. The $O_3$ increase in areas below an $O_3$ decrease has already been reported by Solomon et al. (1985) and they expect a larger effect for lower latitudes, which would agree with our results. Noteworthy is that the area of $O_3$ increase overlaps with the area where $NO_x$ perturbations are close to zero. Additionally, more of the high-energy radiation should reach lower altitudes due to the decrease above the area of $O_3$ increase.

     To conclude, in the EMAC results, either the uncertainties due to annual variability is larger compared to LMDZ-INCA for

420     both $O_3$ and $NO_x$ or the perturbation to background ratio is larger. Additionally, the larger vertical resolution in EMAC could explain the different uncertainty value ($p < 0.05$ for EMAC, $p < 0.001$ for LMDZ-INCA)




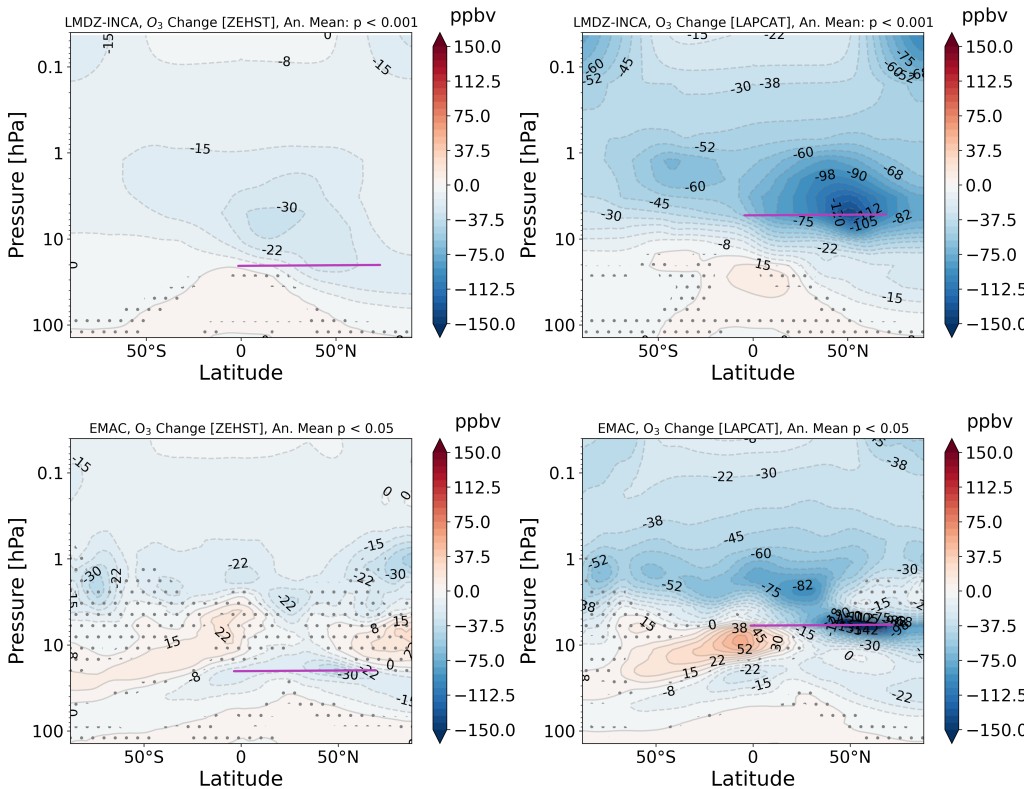

**Figure 10.** Multi-annual mean (2010-2014) of ozone perturbation ($\mathrm{ppbv}$) for the ZEHST scenario (left) and the LAPCAT scenario (right) for both models LMDZ-INCA (top) and EMAC (bottom). Purple lines represent respective cruise altitudes. Hatched areas are characterized by a p-value higher than the threshold indicated in the title of each plot.

The total reduction of $O_3$ is listed in Table 4. In general, results are of the same order of magnitude. Both models show the same trend that the higher flying aircraft fleet has a larger impact on $O_3$ and for both aircraft the perturbation is slightly larger in LMDZ-INCA results.

**Table 4.** Ozone change in percent for each model and each aircraft fleet above the tropopause.

| Scenario | EMAC | LMDZ-INCA |
|----------|---------|-----------|
| ZEHST | -0.068 % | -0.097 % |
| LAPCAT | -0.14 % | -0.17 % |





## 6 Radiation and Climate

The annual radiative impact was calculated with the metric stratosphere adjusted radiative forcing at tropopause level using the equilibrium perturbation of $H_2O$, $O_3$ and $CH_4$. Atmospheric composition changes were used to calculate the RF with both models. The spin-up phase was three months and the averaged result is based on twelve monthly means. The resulting RF for both models and both aircraft fleets are listed in Table 5 with the LAPCAT scenario showing larger values mainly due to the larger water vapour perturbation. The normalized RF per teragram of water vapour perturbation is in good agreement with an average and standard deviation of $0.43 \pm 0.02$ $mW(m^2 * Tg)^{-1}$ and $0.39 \pm 0.02$ $mW(m^2 * Tg)^{-1}$ for EMAC and LMDZ-INCA, respectively. Another measure, the RF per teragram of annual water vapour emission, shows that the normalized RF correlates with altitude and the values are 25-41 % larger for LAPCAT in both model results.

**Table 5.** Radiative forcing per year in $\mathrm{mWm}^{-2}$, per teragram of $H_2O$ perturbation and per teragram of annual water vapour emission (from Table 2), both in $\mathrm{mW(m^2Tg)}^{-1}$, for each scenario and model calculated with atmospheric composition changes of $H_2O$, $O_3$ and $CH_4$.

| Scenario | EMAC $mWm^{-2}$ | LMDZ-INCA $mWm^{-2}$ | EMAC $mW(m^2Tg)^{-1}$ | LMDZ-INCA $mW(m^2Tg)^{-1}$ | EMAC $mW(m^2Tg)^{-1}$ | LMDZ-INCA $mW(m^2Tg)^{-1}$ |
|---|---|---|---|---|---|---|
| ZEHST | 20.95 | 15.42 | 0.44 | 0.40 | 1.52 | 1.12 |
| LAPCAT | 40.31 | 33.49 | 0.41 | 0.37 | 1.90 | 1.58 |

The perturbations of $H_2O$, $O_3$ and $CH_4$ above the meteorological tropopause were used to calculate the RF (approximately 100 hPa at tropical and 300 hPa at polar latitudes). The direct impact of hydrogen perturbation on RF is not significant and their indirect effect, i.e. on $O_3$ and $H_2O$ mixing ratios is included and will be reported in a separate publication in more detail (Pletzer and Grewe, 2022, in prep). In comparison, the contribution to RF of $H_2O$ is largest, followed by $O_3$ and a negative RF due to $CH_4$ reduction. The negative RF of the latter is due to the enhanced methane oxidation and is larger for the LAPCAT scenario where hydroxyl radicals are clearly more active (Fig. 7). The detailed values with short- and longwave contributions are listed in Table 6.

LMDZ-INCA shows a smaller longwave RF (LW RF) and a larger negative shortwave RF (SW RF) for $H_2O$ compared to EMAC. Ozone longwave RF has a negative sign for EMAC and positive for LMDZ-INCA. The differences in magnitude and sign of $O_3$ longwave RF may originate from the varying atmospheric composition changes in EMAC with areas of ozone increase and decrease. This altitude dependency very much affects the contribution to RF as has been shown by Lacis et al. (1990) and Hansen et al. (1997). An additional test shows that the ozone increase in the UTLS compensates the negative longwave forcing due to the high sensitivity to ozone changes and explains the positive longwave value for LMDZ-INCA. Ozone shortwave RF is generally larger by 40-44 % for LMDZ-INCA compared to EMAC. Methane net RF is 1-2 order of magnitude smaller for EMAC compared to LMDZ-INCA. A comparison of change in global methane lifetimes is shown in the appendix. There, methane lifetime change is larger for the LAPCAT compared to the ZEHST scenario for both models and





**Table 6.** Short- and longwave contributions (respectively SW and LW) to radiative forcing (RF) in $\mathrm{mWm}^{-2}$.

| Aircraft | Model | Perturbation | SW RF | LW RF | RF |
|---|---|---|---|---|---|
| ZEHST | LMDZ-INCA | $O_3$ | 3.43 | 0.35 | 3.78 |
| | EMAC | $O_3$ | 2.38 | -0.73 | 1.65 |
| | LMDZ-INCA | $CH_4$ | - | - | -1.38 |
| | EMAC | $CH_4$ | - | - | -0.015 |
| | LMDZ-INCA | $H_2O$ | -2.27 | 15.29 | 13.02 |
| | EMAC | $H_2O$ | -1.59 | 20.90 | 19.32 |
| LAPCAT | LMDZ-INCA | $O_3$ | 5.72 | 1.84 | 7.56 |
| | EMAC | $O_3$ | 4.11 | -0.78 | 3.34 |
| | LMDZ-INCA | $CH_4$ | - | - | -2.47 |
| | EMAC | $CH_4$ | - | - | -0.046 |
| | LMDZ-INCA | $H_2O$ | -4.25 | 32.65 | 28.40 |
| | EMAC | $H_2O$ | -2.27 | 39.90 | 37.01 |

the methane lifetime is in general less for EMAC compared to LMDZ-INCA for both ZEHST and LAPCAT scenarios by a factor of three and two, respectively. To conclude, the reason for the different magnitude in methane net RF is not fully clear and should originate from the RF calculation. However, the contribution of methane to RF is small in both models compared to $H_2O$ and $O_3$ and should not affect our results.

For the largest contribution to RF, the $H_2O$ perturbation, we have performed a comparison to other radiation calculations.
The performance test was done like in Myhre et al. (2009). That means we calculated the impact on RF by an increase of water vapour mixing ratio from 3.0 to 3.7 ppmv above the tropopause. The LMDZ-INCA result is 0.18 $Wm^{-2}$, which is below the mean of Myhre et al. (mean 0.25 $Wm^{-2}$, range 0.16 – 0.38 $Wm^{-2}$), while the EMAC result is larger than the mean with 0.28 $Wm^{-2}$. Both models are in the range of different models presented by Myhre et al., with LMDZ-INCA at the lower and EMAC in the mid-upper range.

Perturbations at tropospheric altitudes were neglected mainly due to the large variability of water vapour, either with a reset to reference water vapour during the perturbation simulations (LMDZ-INCA) or due to exclusion in the RF calculations (EMAC). Hence, for EMAC the $H_2O$ perturbations at tropospheric altitudes are not zero and therefore will contribute to the RF calculations. However, the upper tropospheric water vapour perturbations have a very large variability, since the 5 % confidence intervals for the mean are ±106 % and ±33 % for the ZEHST and LAPCAT scenarios, respectively. In comparison, for $H_2O$
perturbations above the tropopause, variability is significantly smaller with ±3.1 % and ±1.8 % for the ZEHST and LAPCAT scenarios, respectively. Hence, water vapour perturbation in the upper troposphere could significantly contribute to RF, but the associated error due to the large variability is significantly larger. Nonetheless, we calculated the RF with EMAC including the





tropospheric perturbations for comparison (see Table 7). We found that the RF increases by approximately 51 to 63 % when the upper tropospheric perturbations are included.

**Table 7.** Short- and longwave contributions to radiative forcing in $\mathrm{mWm^{-2}}$ for EMAC, including upper tropospheric perturbation of water vapour, ozone and methane, and the related error potential of approximately 51 to 63 % due to the integration of tropospheric perturbations.

| Aircraft | Perturbation | SW RF | LW RF | RF |
|----------|--------------|-------|-------|-----|
| ZEHST | $H_2O$ | 0.07 | 31.43 | 31.50 |
| LAPCAT | $H_2O$ | -0.44 | 56.18 | 55.74 |

## 7 Discussion

### 7.1 Atmospheric Composition Changes

Here, we focus on the comparison to the publication by Kinnison et al. (2020). They estimate the ozone, water vapour, $\mathrm{NO_x}$ and $\mathrm{HO_x}$ perturbations by a fleet of hypersonic aircraft in independent scenarios where aircraft, powered with conventional fuel, fly at 30 km and 40 km, using the coupled chemistry-climate model WACCM (Whole Atmosphere Community Climate Model). Similar to our setup they look at atmospheric conditions for the year 2050, however the averaged annual results are based on one year, while ours represent the mean over four years. In turn, a larger deviation from the long-term mean could be expected for their results. The total annual emission in their setup amounts to 58.3 Tg of $H_2O$ and 0.94 Tg of $NO_2$, which is approximately three to four times larger for the former, and 30 to 47 times larger for the latter, compared to the annual values of the HIKARI data.

In agreement with our result, the $H_2O$ perturbation for the higher flying aircraft fleet is significantly larger and the perturbation patterns agree very well with the maximum perturbation being at midlatitudes at the northern hemisphere and at the cruise altitude. In their case the total emission per trace gas is the same for both altitudes, which is not the case in our study, since the emissions in HIKARI are based on two different aircraft designs. Total values of mass perturbation were not published for $H_2O$ and thus cannot be compared with ours. We presented the (photo-)chemical lifetime in Fig. 6. Here, the agreement of latitudinal and altitude features with Kinnison et al. (2020, Fig. 3) is very good in general. There are differences in magnitude from 50 km upward, increasing with altitude where photolysis is dominant. If this originates from WACCM covering more of the atmosphere compared to our models, and thus being more accurate at 50 km upwards, is an open question. However, the stratosphere is well represented in the models used here, which is most important for the topic of our study, and the equilibrium mass perturbation at mesospheric altitudes is insignificant in comparison to stratospheric altitudes.

The calculated ozone sensitivity with -4.2 % / $\mathrm{Tg\,NO_2}$ and -2.4 % / $\mathrm{Tg\,NO_2}$ for 30 km and 40 km altitude (Kinnison et al., 2020, Table 2) is of the same order of magnitude compared to our results with -3.4 % / $\mathrm{Tg\,NO_2}$ and -4.5 % / $\mathrm{Tg\,NO_2}$




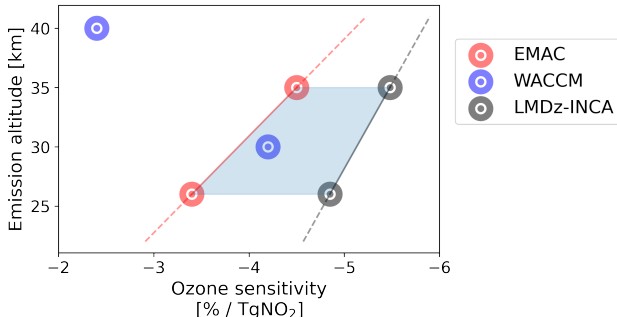

**Figure 11.** Ozone sensitivity from three different models WACCM, EMAC and LMDZ-INCA dependent on altitude. Results from this work were calculated using values from Table 2 and Table 4. Results for WACCM were published by Kinnison et al. (2020, Table 2). The shaded area highlights the good agreement of LMDZ-INCA and EMAC results with WACCM results for the region from approximately 26-35 km altitude.

for EMAC and -4.9 % / Tg $NO_2$ and -5.5 % / Tg $NO_2$ for LMDZ-INCA at 26 km and 35 km altitude, respectively. Figure 11 shows both results from our study and their study. Our results have a positive correlation between ozone sensitivity and altitude. This might point to a maximum of absolute ozone sensitivity at around 35 km altitude, since the absolute value for
WACCM at 40 km altitude is already much smaller and the models seem to agree very well for the region between 26 and 35 km. The assumed tropical maximum of ozone mixing ratio of 31 km overlaps with this region and is very close to the maximum value. However, the altitude of emission often does differ from perturbation maxima of $NO_x$ and $O_3$. Be aware that we are only presenting two data points per model and to come to conclusions regarding the largest value of ozone sensitivity might be inaccurate. Additionally, there are some differences between the setups, e.g. Kinnison et al. estimate ozone sensitivity based on
$NO_x$ perturbations only and with a larger amount, while we do look at the combined effects by $NO_x$, $H_2O$ and $H_2$ emission. Furthermore, the HIKARI data includes a vertical distribution of emission, in which take off and landing are present, and the fleet comprises not only hypersonic but subsonic aircraft as well, while they inject the emission in a single layer. The effect of tropospheric water vapour emission on SWV is negligible due to the tropical tropopause coldpoint, but $NO_x$ emitted in the tropical troposphere may be transported to stratospheric altitudes and increase the uncertainty of the comparison. Note that in
another set of EMAC simulations, from a forthcoming publication (Pletzer and Grewe, 2022, in prep), where $NO_x$ is emitted in a single layer, we see an approximately equal ozone sensitivity at 30 km and 38 km for tropical and midlatitudinal regions, while for northern polar regions the lower altitude has a sensitivity nearly twice as much as the higher altitude and shows a negative correlation very similar to Kinnison et al. (2020).

## 7.2 Comparison to the Climate Impact of Other Aircraft Designs

For a better comparison of hypersonic to sub- and supersonic aircraft we included Fig. 12. It shows an enhancement factor, i.e. the ratio of the climate impact of a specific aircraft compared to a conventional subsonic aircraft, depending on altitude. The numbers shown there were calculated using the climate response model AirClim (Grewe and Stenke, 2008; Dahlmann et al.,





2016). The comparison is based on results from this study and Grewe et al. (2007, 2010); Grewe (2021). While subsonic is the reference case with a value of 1, supersonic aircraft show a climate impact that is increasing with altitude. The hypersonic
aircraft ZEHST follows that trend with an enhancement factor of approximately 20 (near-surface temperature change or RF normalized to revenue passenger kilometers). However, the enhancement factor of the second hypersonic aircraft LAPCAT (PREPHA) is less than 10 due to its higher passenger capacity. Hence, a larger aircraft size, i.e. larger passenger number, is clearly a promising design option to reduce the climate impact per passenger and can compensate the climate impact due to higher cruise altitudes. The error estimate (blue shaded area) includes the tropospheric region with its very large variability (see
page 27). Blue squares represent an aircraft of type LAPCAT MR3, developed in the STRATOFLY project, for four different altitudes and based on a single trajectory (Viola et al., 2021). There, the increasing climate impact with altitude is clearly visible.

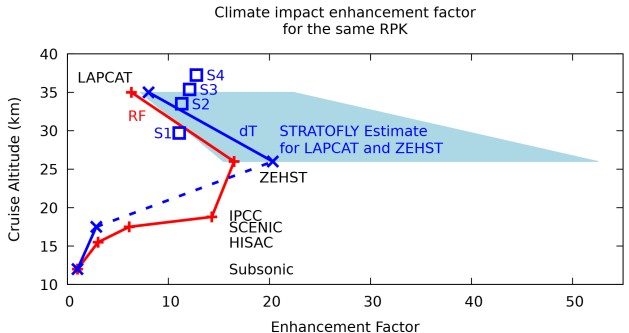

**Figure 12.** Radiative forcing (red) 50 years after entry into service (EIS) of a fleet of the respective aircraft and the near-surface temperature change (blue) based on the HIKARI project results. RPK (unit pax-km) refers to the revenue passenger kilometers, i.e. total kilometers travelled by all passengers on an aircraft or on a fleet of aircraft. The shaded area shows an uncertainty range from this work. This figure is taken from Grewe (2021) and includes the values of four STRATOFLY MR3 versions for different altitudes (S1 to S4) based on data published by Viola et al. (2021, Table 2).

## 7.3  Climate Impact of Hypersonic Aircraft

Estimates on the climate impact of hypersonic aircraft barely exist. A recent estimate was published by Ingenito (2018).
He approximates the climate impact by a fleet of hypersonic aircraft (type LAPCAT II MR2.4, 2015) based on water vapour perturbation only. In his study a fleet of 200 hypersonic aircraft fly from Brussels to Sydney 365 days a year and emit 376 $\mathrm{Tg}$ of water vapour, which results in a water vapour perturbation that increases surface temperature by 100 $\mathrm{mK}$. We want to mention that the estimate is based on a correlation of an increase in global atmospheric water vapour and near-surface temperature from a third publication and the whole calculation can be described as a 1D box model. For comparison to our results in Fig.
12, we normalize the change in surface temperature with passenger kilometers (pax-km). Therefore, we assume a distance of 16,367 km between Brussels and Sydney and a passenger capacity of 300 (LAPCAT II) and obtain an enhancement factor of



$396\ 10^{-12}\ \mathrm{mK(pax - km)^{-1}}$. This equals a near-surface temperature change due to hypersonic aircraft 61 times as much as subsonic aircraft. Compared to the enhancement factor in Fig. 12, this value is outside of the uncertainty range of our study. The upper limit of the uncertainty range is equal to RF calculations with composition changes including upper tropospheric water
vapour. These were neglected in the main calculations due to the large variability and the focus on stratospheric perturbation of trace gases.

We mentioned the significant contribution of upper tropospheric water vapour to RF in our model simulations with EMAC and want to elaborate some more, as this is important for comparison. In general, the lifetime of water vapour is comparably short at tropospheric altitudes (spreading from hours to approximately six months, Fig. 6a in Grewe and Stenke (2008)).
However, in our study the main emission is at midlatitudes, where water vapour lifetime is between 7 days and one month according to the reference. We did not test how well the tropospheric perturbation is represented in our model, since we focused on stratospheric perturbations in the EMAC setup. In EMAC simulations, the tropospheric water vapour was not reset nor nudged to ECMWF data, in contrast to the LMDZ-INCA simulations. Therefore, the variability introduces a large error range in the upper tropospheric water vapour results compared to the results for the stratosphere in EMAC.

## 8  Summary

In this study we calculated the radiative forcing and the climate impact of two different hypersonic aircraft designs, both fueled with liquid hydrogen. The difference in cruise altitude (26 km and 35 km) results in significant differences in atmospheric perturbations, perturbation lifetime and in turn climate impact. Clearly, water vapour is the largest contributor for the latter. We find an efficient (photo-)chemical destruction of $H_2O$ at higher altitudes, as expected based on theory (e.g. Fig. 5.21, p. 231
Brasseur and Solomon, 1984). But we did not see a smaller $H_2O$ perturbation at the higher emission altitude, which agrees with the tendency in the study by Kinnison et al. (2020). Our analysis shows for the first time that a recombination to $H_2O$ overcompensates the (photo-)chemical destruction of emitted $H_2O$, which results in a longer $H_2O$ perturbation lifetime at the higher altitude. This may change at even higher altitudes, where water vapour lifetime decreases and, more importantly, the lifetime of $HO_x$ and $H_2$ increases substantially (Brasseur and Solomon, 1984, Fig. 5.28, p. 242). Whether the recombination
of emitted water vapour is affected has not been tested at these altitudes. The finding of longer $H_2O$ perturbation lifetime and recombination contributes to the understanding of chemistry at stratospheric and lower mesospheric altitudes. For ozone, we report an overall depletion of the ozone layer, with a decrease at middle-to-upper stratospheric altitudes and an increase in lower stratospheric altitudes. This results in radiative warming for both aircraft, with a larger effect for the higher flying aircraft. The radiative impact of increased hydrogen is not significant compared to the other contributors and has a comparably
small indirect effect by contributing to atmospheric perturbations of $CH_4$, $H_2O$ and $O_3$. This will be addressed in more detail in a follow-up publication. The results can be deemed robust due to the usage of two high-end chemistry climate models and the performance validation of radiative forcing due to stratospheric water vapour perturbation. To conclude briefly, the impact on climate of aircraft emitting water vapour and flying above the tropopause increases very much with altitude. This is clearly shown by the increase of water vapour perturbation lifetime and normalized radiative forcing with altitude. Due to larger fuel



consumption with higher speed at high cruise altitudes on one hand, and the atmospheric conditions at these cruise altitudes (recombination, lifetime of $H_2O$) on the other hand, hypersonic aircraft have a considerable larger climate impact than subsonic and supersonic aircraft.



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




**Appendix A**

The applied EMAC model setup comprised the submodels AEROPT (AERosol OPTical properties), AIRSEA, CH4, CLOUD,
CLOUDOPT (CLOUD OPTical properties), CONTRAIL, CONVECT (CONVECTion), CVTRANS (Convective Tracer TRANS-
port), DRADON (Decay RADioactive ONline), DDEP, E5VDIFF (ECHAM5 Vertical DIFFusion), GWAVE (Gravity WAVE),
H2OEMIS, JVAL (J VALues), LNOX, MECCA, MSBM, O3ORIG (O3 ORIGin), OFFEMIS, ONEMIS, ORBIT, OROGW
(OROgraphic Gravity Waves), PTRAC (Passive TRACers), QBO (Quasi Biannual Oscillation), RAD (RADiation), S4D (Sam-
pling in 4 Dimensions), SATSIMS (Satellites Simulator), SCALC (Simple CALCulations), SCAV, SCOUT (Stationary Column
OUTput), SEDI, SORBIT (Satellite ORBITs), SURFACE, TBUDGET, TENDENCY, TNUDGE (Tracer NUDG(E)ing), TR-
EXP (Tracer Release EXperiments from Point sources), TROPOP (TROPOPause) and VISO (Vertically layered iso-surfaces
and maps) (Jöckel et al., 2006, 2010; Roeckner et al., 2006). Further information is available on the MESSy homepage
https://www.messy-interface.org/.

**Table A1.** Change in global methane lifetime for LMDZ-INCA and EMAC and the ZEHST and LAPCAT scenarios.

| Scenario | EMAC | LMDZ-INCA |
|---|---|---|
| ZEHST | -0.03 % | -0.09 % |
| LAPCAT | -0.09 % | -0.16 % |





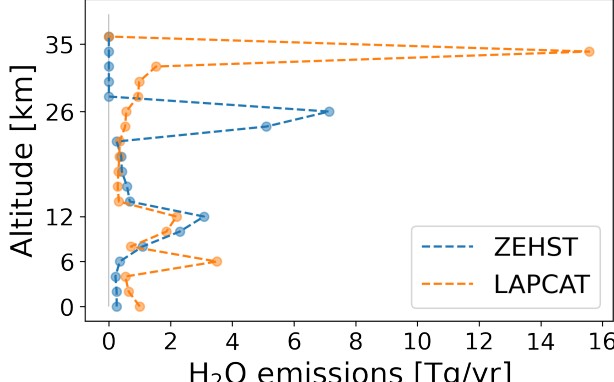

**Figure A1.** Vertical distribution of annual $H_2O$ emission for ZEHST and LAPCAT. The peak at 6 km altitude comes from hypersonic boost of the LAPCAT aircraft. The vertical distribution was aggregated for a better comparability (with a 2 km bin size). For the total amount of emission, see Table 2.




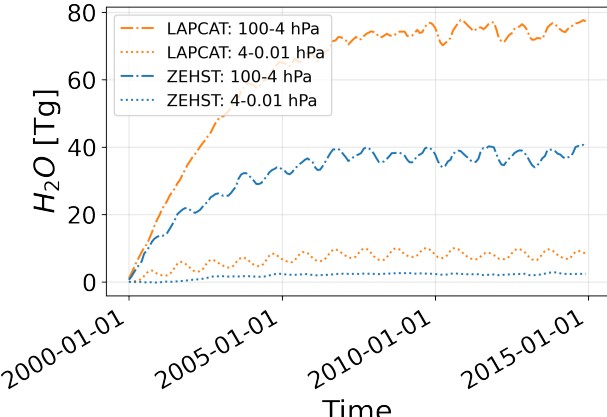

**Figure A2.** Timeline plot of accumulated trace gas $H_2O$ in teragram, based on monthly mean values. Blue lines represent scenario ZEHST and orange lines represent scenario LAPCAT. Shown is the $H_2O$ perturbation over time above 4 hPa (dotted) and from 4-100 hPa (solid-dotted).





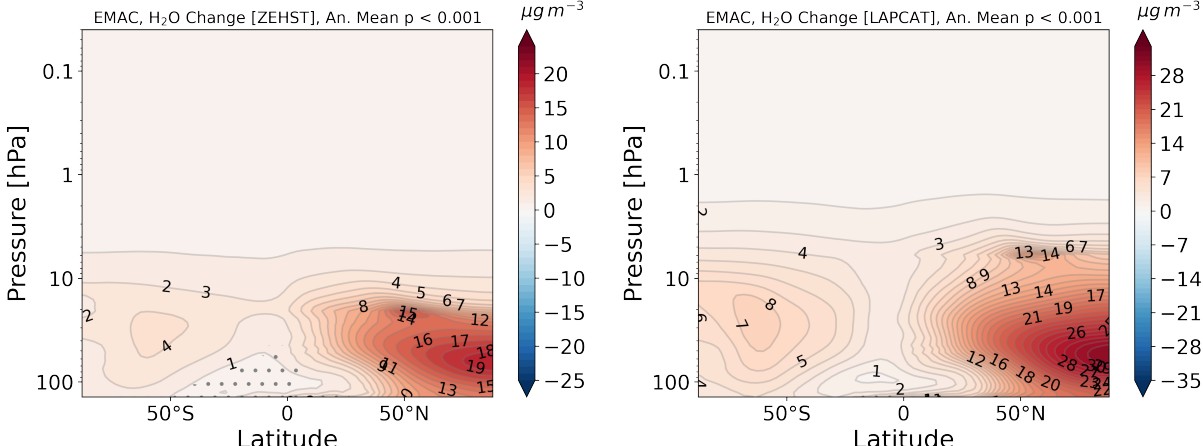

**Figure A3.** Multi-annual mean (2010-2014) of $H_2O$-perturbation [µg m$^{-3}$] for the ZEHST scenario (a) and the LAPCAT scenario (b). Dotted areas represent probabilities larger than 0.1 % for data not to be significant (standard t-test). Horizontal lines represent respective cruise altitudes.



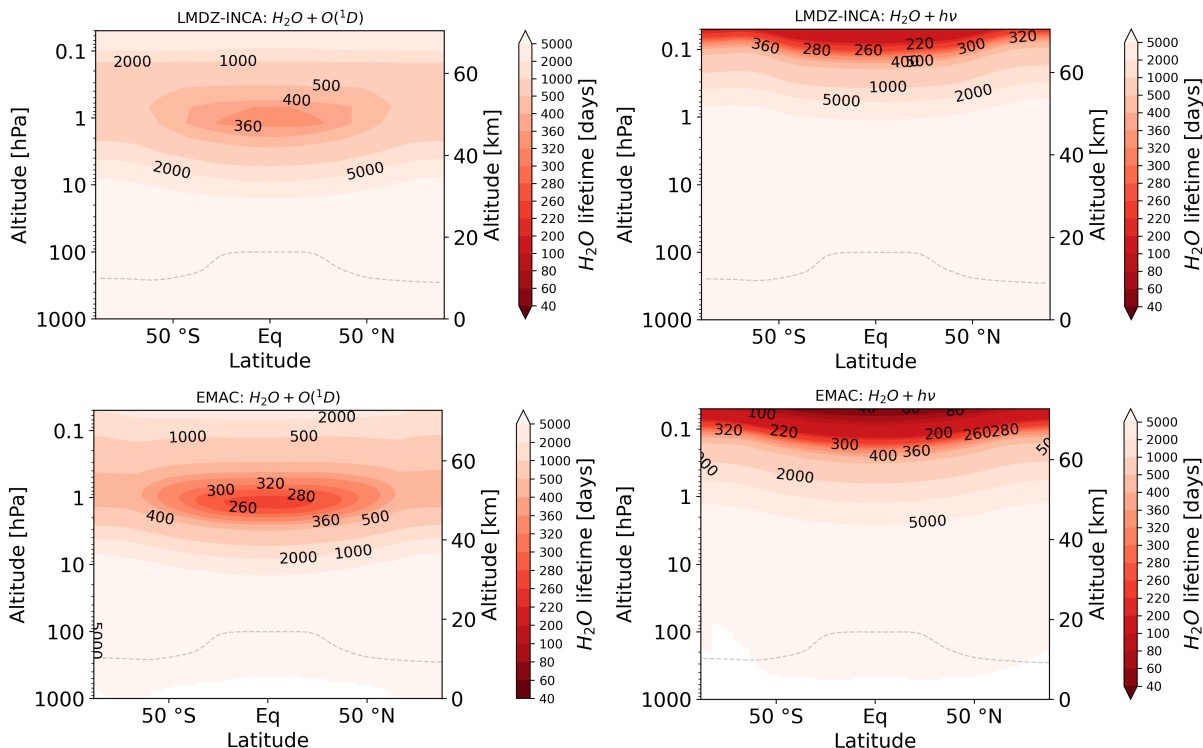

**Figure A4.** Zonal mean (photo-)chemical $H_2O$ lifetime in days for photolysis (right column) and reaction with $O(^1D)$ (left column) for LMDZ-INCA (upper row) and EMAC (lower row).





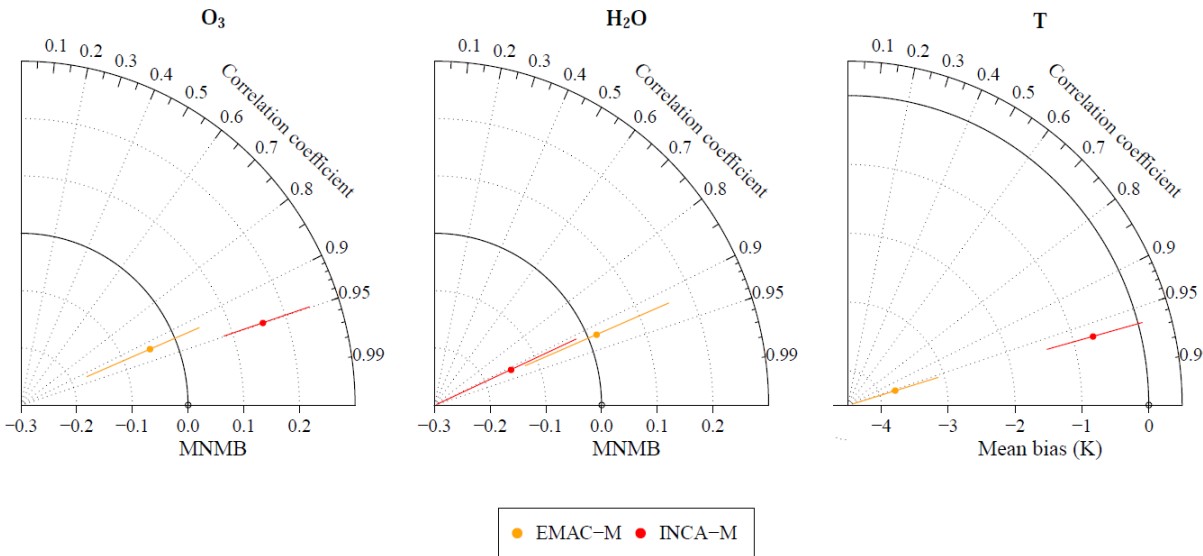

**Figure A5.** Same as Fig. 4 for winter.





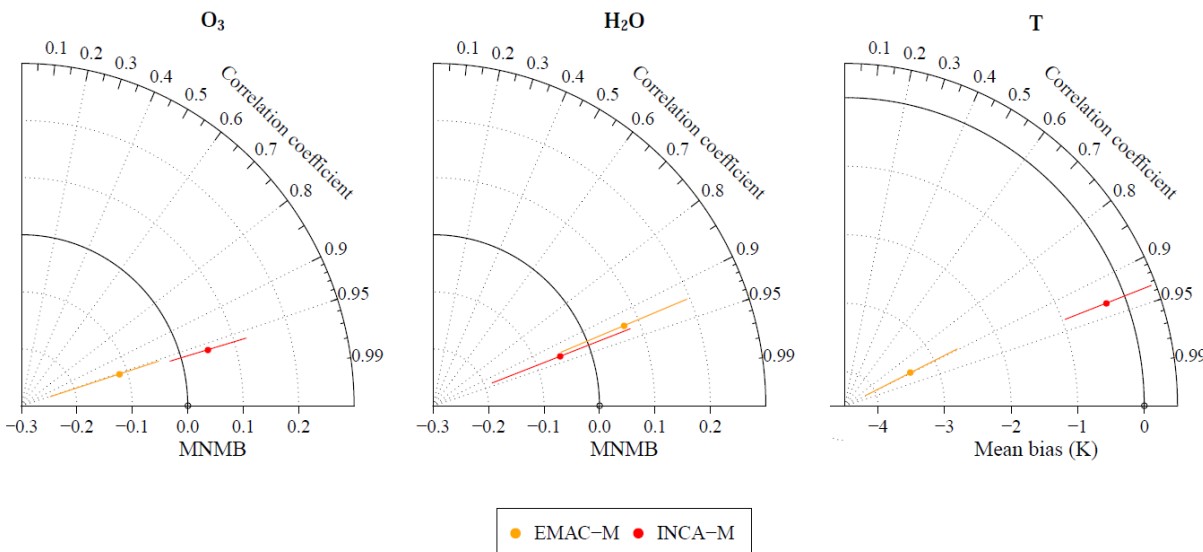

**Figure A6.** Same as Fig. 4 for spring.





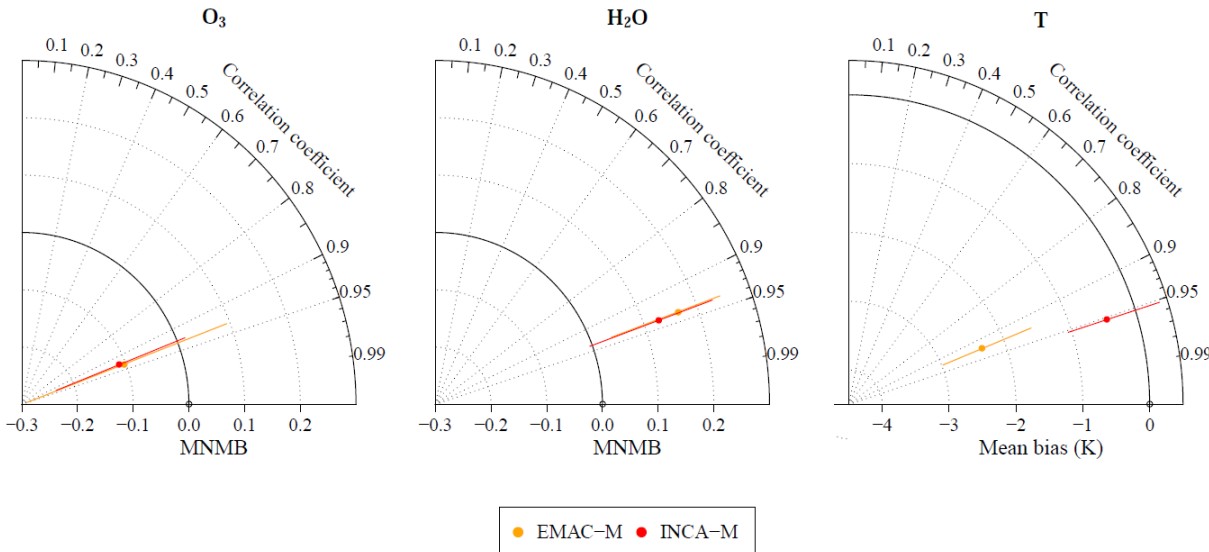

**Figure A7.** Same as Fig. 4 for summer.





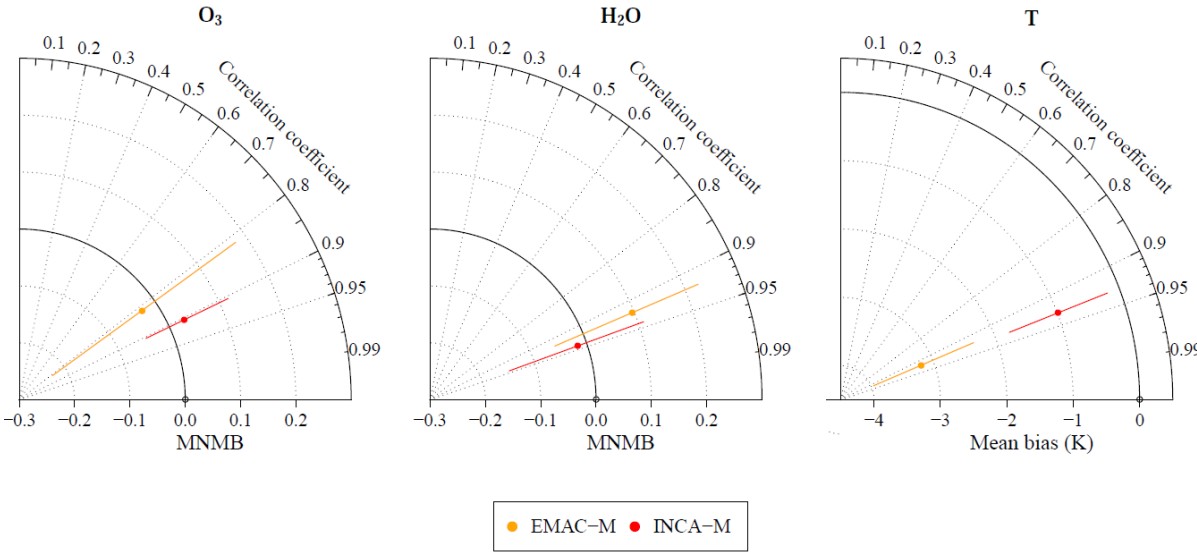

**Figure A8.** Same as Fig. 4 for autumn.





**Figure A9.** Annual mean climatologies for ozone, water vapour and temperature (from top to bottom) averaged over 2000–2014, and vertically through the cruise levels. The first and second columns represent the climatologies seen by the IAGOS observations, projected onto the EMAC grid and onto the INCA grid respectively, the latter being interpolated onto the EMAC grid afterward. The third and fourth columns show the biases between the reference simulation for each model and its corresponding gridded IAGOS product. For each grid cell, the normalized bias for the mixing ratios is calculated relatively to the average between observations and the model simulation.







**Figure A10.** Same as Fig. A9 for boreal winter.





**Figure A11.** Same as Fig. A9 for boreal spring.




**Figure A12.** Same as Fig. A9 for boreal summer.





**Figure A13.** Same as Fig. A9 for boreal autumn.



**Code availability**

The Modular Earth Submodel System (MESSy) is continuously further developed and applied by a consortium of institutions. The usage of MESSy and access to the source code is licenced to all affiliates of institutions which are members of the MESSy Consortium. Institutions can become a member of the MESSy Consortium by signing the MESSy Memorandum of Understanding. More information can be found on the MESSy Consortium Website (http://www.messy-interface.org). The submodel H2OEMIS (water vapour emissions) presented here has been implemented in MESSy version 2.54.0 and is available 790 in the official release 2.55.0. The LMDZ-INCA global model is part of the Institut Pierre Simon Laplace (IPSL) Climate Modelling Center Coupled Model. The documentation on the code and the code itself can be found at https://cmc.ipsl.fr/ipsl-climate-models/ipsl-cm6/. The distribution of the IAGOS data onto the EMAC and LMDZ-INCA grids is based on an updated version of the Interpol-IAGOS software, available at https://doi.org/10.25326/81 (Cohen et al., 2021b).

**Data availability**

To access datasets of LMDZ-INCA results please contact Didier Hauglustaine. For access to datasets of EMAC results please contact Johannes Pletzer. The IAGOS data set can be found at http://doi.org/10.25326/20 (Petzold et al., 2015) and more precisely, the IAGOS time series used in this study are available at http://doi.org/10.25326/06 (Bundke et al.).

*Video supplement.* Two video supplements are published with a doi via zenodo.org. The first supplement shows the time development of SWV and integrated $H_2O$ perturbation by LAPCAT aircraft at 32-38 km altitude (http://doi.org/10.5281/zenodo.4455592) over a time period 800 of fourteen years (2000-2014). The second supplement shows the transport of $H_2O$ emission by LAPCAT aircraft in the first month of airfleet operation at stratospheric altitudes. The presentation is a world map view (http://doi.org/10.5281/zenodo.4475334).

*Author contributions.* Johannes Pletzer set-up the EMAC model simulations, analysed and post-processed the EMAC model results. Didier Hauglustaine set-up the LMDZ-INCA model simulations, analysed and post-processed the LMDZ-INCA model results, which Johannes Pletzer plotted for a consistent presentation. Yann Cohen contributed with the IAGOS observation validation of both models. Patrick Jöckel 805 helped with the choice of the initial EMAC setup and with the writing of the H2OEMIS submodel. Volker Grewe was involved in the discussion of the results and supported the writing of the document. The manuscript was written by Johannes Pletzer, then reviewed by the authors and approved by all the authors.

*Competing interests.* The authors declare that they have no conflict of interest.





*Acknowledgements.* The authors gratefully acknowledge D. Kinnison and G. Brasseur for helpful discussions on their results.


MOZAIC/CARIBIC/IAGOS data were created with support from the European Commission, national agencies in Germany (BMBF), France (MESR), and the UK (NERC), and the IAGOS member institutions (http://www.iagos.org/partners). The participating airlines (Lufthansa, Air France, Austrian, China Airlines, Iberia, Cathay Pacific, Air Namibia, Sabena) supported IAGOS by carrying the measurement equipment free of charge since 1994. The data are available at http://www.iagos.fr thanks to additional support from AERIS.


The model simulations have been performed at the German Climate Computing Centre (DKRZ). The resources for the simulations were offered by the Bundesministerium für Bildung und Forschung (BMBF). The LMDZ-INCA simulations were performed using HPC resources from GENCI (Grand Equipement National de Calcul Intensif).

The H2020 STRATOFLY Project and the H2020 MORE&LESS project have received funding from the European Union's Horizon 2020 research and innovation program under Grant Agreement No. 769246 and 101006856, respectively.

The main part of post processing, data analysis and plotting has been done using the module *xarray* (Hoyer and Joseph, 2017, v0.20.1) and *matplotlib*.


For interpolating hybrid model level to pressure level we used the *interp hybrid to pressure* function from *geocat.comp* (Visualization and Analysis Systems Technologies, 2021).