# Peer review of "The Climate Impact of Hydrogen Powered Hypersonic Transport"

_EGUsphere, 2022_

## Referee Comment (RC1)

**Review of "The Climate Impact of Hypersonic Transport"**
**by Pletzer et al.**

**General Comments**

The paper by Pletzer et al. provides a modeling study of the potential climate and ozone impacts due to emissions from Hypersonic Transport. The study is motivated by the growing contribution from aviation to climate change and also increasing desire to reduce air travel time. The present paper evaluated two hypersonic aircraft designs using two 3D chemistry-climate models, providing an assessment on climate and ozone impact for flying these aircraft. The present paper shows for the first time that a net chemical production at high altitudes leading to an increase of $H_2O$ perturbation lifetime and $H_2O$ concentration.

This is a very interesting and well-written paper that is a significant contribution that merits prompt publication. I only have a few minor suggestions for the authors to consider (below). What I find 'missing' is some contextual information. Also, I suggest that the limitations of the models need to be discussed more explicitly. I have detailed some of this below that might easily rectify this for the authors to consider.

**Specific Comments**

In the methods and simulations section, it is still not clear to me what nudging is applied in the models? I am assuming U wind, V wind and surface pressure? Is there anything else? Is it nudged all the way from surface to the model top? I assume the ERA-Interim meteorology fields first get interpolated to model grid (both horizontally and vertically) offline before nudged to the models? Please explain/clarify in the paper.

L151
"The respective relaxation times are listed in 1."
The respective relaxation times are listed in Table 1?

L190 and L238
Two models seem to have large difference in derived lightning NOx – 0.2 TgN/yr and 5.5 TgN/yr in EMAC and LMDZ-INCA respectively. Why is that? How do these two models derived the lightning NOx. How does this large difference can potentially influence the results here, if any? Hope the authors could elaborate more on this.

L194
What are the vertical resolutions (especially in the upper troposphere and stratosphere) in the two models? What is the vertical resolution of the HIKARI aircraft emission inventory? What is the underline assumption of the aircraft emission spread throughout the model grid? What is the role of model diffusivity in the distribution of emissions? What is the limitation? More discussion on this aspect would be helpful.

L240 to L250
The model uses the meteorology data from the time period 2000-2014 to simulate climate impact in 2050-2064. One can argue that using the meteorology fields between 2000 to 2014 doesn't account for the dynamics changes (e.g., temperature and circulation) at the upper troposphere and lower stratosphere owning to climate change. For example, previous studies have shown that the Brewer–Dobson circulation has been strengthening in response to climate change (e.g., Butchart et al., 2006; Shepherd et al., 2011). How much difference can it make to the resulting effects?

Butchart, N., Scaife, A. A., Bourqui, M., De Grandpré, J., Hare, S. H. E., Kettleborough, J., ... & Sigmond, M. (2006). Simulations of anthropogenic change in the strength of the Brewer–Dobson circulation. *Climate Dynamics*, 27(7), 727-741. DOI 10.1007/s00382-006-0162-4

Shepherd, T. G., & McLandress, C. (2011). A robust mechanism for strengthening of the Brewer–Dobson circulation in response to climate change: Critical-layer control of subtropical wave breaking. *Journal of the Atmospheric Sciences*, 68(4), 784-797. https://doi.org/10.1175/2010JAS3608.1

Figure 1 shows the vertical sum of H2O emissions over all vertical levels. How different the figure would be if one only plotted the emissions at cruise altitudes?

L295
States that "Most of the emitted trace gases are transported to tropospheric altitudes". I am curious how much emitted H2O is chemically destroyed in the stratosphere? Have the authors explored the ratio of emitted stratospheric trace gases transported to tropospheric altitudes VS chemical destruction in the stratosphere? Which is the major contributor (transport or chemistry) to the stratospheric H2O sink here? It would be very interesting to see.

Table 3
It would be interesting to see how much percentage of total H2O perturbation stay in the atmosphere (above the tropopause) when it reaches equilibrium (e.g., Perturb. LAPCAT/ total LAPCAT emission *100% in two models)

Table 4
shows that LMDZ-INCA calculates a higher ozone destruction than EMAC. The difference can be about 42% for lower flying aircraft (ZEHST). What factors can be mainly contributed to the difference derived in two models? Could the authors add a few sentences to discuss about this?

L476
"over four year"
over 5 years? 2010-2014

A recent study by Zhang et al., 2021 calculated the stratospheric adjusted radiative forcing of water vapour and ozone perturbation at difference cruise altitudes. How does the water vapor and ozone radiative forcing impact in this study compared to their study? Scaling by per unit H2O emission in $mW(m^2Tg)^{-1}$?

Zhang, J., Wuebbles, D., Kinnison, D., & Baughcum, S. L. (2021). Stratospheric ozone and climate forcing sensitivity to cruise altitudes for fleets of potential supersonic transport aircraft. *Journal of Geophysical Research: Atmospheres*, 126(16), e2021JD034971. https://doi.org/10.1029/2021JD034971

L563
"To conclude briefly, the impact on climate of aircraft emitting water vapour and flying above the tropopause increases very much with altitude." Agree with the authors: this conclusion is indeed valid for climate impact of water vapour emission with altitude; however, it might not be the case for ozone perturbation. Zhang et al study showed that there is an inflection point where the radiative forcing of ozone perturbation changes from positive to negative with increasing the flying altitude.

In Kinnison et al. 2020 study, they have showed in figure 1 that the natural stratospheric photochemical production of water vapor resulting from the oxidation of methane is 60 Tg (H2O)/year, which seems to be much higher than the value (less than 14.6 Tg/yr) presented in Figure 7 and Figure 8 in this study. Could the authors explain a little why there is such a big difference?

The simulations assume an atmosphere without volcanic eruptions, but volcanic eruptions will happen in the time frame of an employment of hypersonic transport. What are possible effects of aerosol particles and water vapour of volcanic origin? How would this effect change the picture put forward here? Perhaps nothing can be said, but then this point should be clearer as a limitation of the study.

L800
"over a time period 800 of fourteen years (2000-2014)"
over a time period 800 of fifteen years (2000-2014).

L812
https://www.iagos.org/partners
The link doesn't work for me

---

## Referee Comment (RC3)

**Review of "The Climate Impact of Hypersonic Transport "**

by Johannes Pletzer et al.

**General remarks**

This paper analyses the Climate Impact of Hypersonic Transport. Very few studies in this direction are available. Therefore such studies are needed and such results certainly constitute an important contribution to ACP. It is also good that two independent models are employed (and the results compared) as specific model features could impact the results of such model studies.

However, as the paper stands, I do not think it is very useful. My most important criticism is that the message (result) of the paper is not clear (see also below). Some issues (e.g. contrail formation by subsonic planes) seem to be overlooked and it is not clear to the reader how the "impact" is quantified and measured. I am also not convinced about the "first time that a recombination to $H_2O$" is found (see equation 1), but such issues should be clearly worked out, not just mentioned in passing. (I note that the key reaction, equation 1, if I understood correctly) is not even explicitly mentioned in the manuscript.

I am sorry for sounding so negative; I think a lot of work on the manuscript is required, but the topic is of great importance.

**Comments**

**Message of this paper**

It is not clear to me what the main message of the paper is. To me it seems that the question is the radiative forcing induced by a fleet of hypersonic planes because of the emissions of $H_2O$ they cause in the stratosphere. A lot of the discussion is along this argument. But then the abstract talks about depletion of the ozone layer (l. 14, but is this depletion in column ozone?) without addressing the processes (is the depletion caused by $NO_x$ or

by $HO_x$ or both)? Is the depletion relevant because of UV issues (the ozone reductions seem small) or because of radiative forcing?

The climate impact is measured relative to subsonic aircraft (l. 16/17), but what quantity is used to calculate the relative impact? Is it radiative forcing? This should be clear from the abstract. Assuming it is radiative forcing does this forcing only consider the impact of $CO_2$ emissions by conventional subsonic aircraft? Such aircraft cause contrails (ice particles) which have a potential impact on radiative forcing (e.g., Kärcher, 1996) – has this effect been considered?

Further, there could be ways to manufacture carbon neutral kerosene like fuels (for subsonic aircraft as well as for supersonic and hypersonic aircraft) and of course a subsonic aircraft in the future which is fuelled by (green) liquid hydrogen is not unthinkable. I understand that this study cannot discuss all of these possibilities, but by making a particular choice, it rans into the danger of giving a biased comparison. And further (see below and the discussion in the manuscript) the lifetimes of radiative forcing of emissions of $H_2O$ and $CO_2$ to the atmosphere are different (and depend very much on altitude in case of $H_2O$).

**Carbon cycle**

It is stated in the paper that "the $CO_2$ perturbation originating from fossil fuel is subject to a large variety of sinks with different lifetimes. In general, the range is approximated with 2-20 centuries, where most of the $CO_2$ climate impact is taken up by ocean and biosphere sinks and 20-35 % remain in the atmosphere for longer time ..."

First, it should be noted that the $CO_2$ that remains in the atmosphere can only be really taken out of the system by sedimentation of carbon containing material to the ocean sediments on timescales much longer than centuries (100 000 years) and, second, the ocean uptake depends on the ocean circulation and ocean water chemistry (which is in the order of perhaps 5000 years) (Archer and Brovkin, 2008).

**Water vapour as a greenhouse gas**

I think we all agree that water vapour is the most important greenhouse gas; it accounts for about half of the present day greenhouse effect and is

the most important gaseous source of infrared opacity in the atmosphere. Tropospheric water vapour warms the climate and so does lower stratospheric water vapour (Solomon et al., 2010; Riese et al., 2012). However, it seems to be that an underlying assumption throughout the paper that stratospheric $H_2O$ always warms. However, at higher stratospheric altitudes and especially in the tropics a unit mass increase in $H_2O$ cools the climate (Riese et al., 2012). The authors might not agree with the results by Riese et al. (2012) but this aspect of heating and cooling should be discussed in the paper.

Overall, I suggest that the background balance of water vapour throughout the stratosphere is considered (e.g., LeTexier et al., 1988; Brasseur and Solomon, 2005; Poshyvailo et al., 2018, and references therein), on top of which the impact of perturbations by the proposed fleet of hypersonic planes can be assessed.

**Water vapour production in the stratosphere**

The authors state in the abstract that "$H_2O$ depletion at high altitudes is overcompensated by a recombination of hydroxyl radicals to $H_2O$ ..." and later in the paper state: "Opposite to the expected removal of $H_2O$ emissions, we found a before unknown net-recombination of $H_2O$" (l. 391).

I think that they are referring to the reaction

$$HO_2 + OH \longrightarrow H_2O + O_2 \tag{1}$$

First, given that this seems to be a major issue for the paper I suggest explicitly stating the reaction. Second, the water vapour production in eq. 1 is known (see, e.g., eq. 5.105 in Brasseur and Solomon, 2005). So I am not sure what "before unknown" means here. Finally, I agree with the authors that the lifetime of water vapour decreases with altitude, but of course water vapour concentrations are a balance of loss and production terms (like it is the case for other species), so I am also not sure about "expected removal of $H_2O$ emissions".

**The debate on supersonic transport**

I realise that this is not a historical paper and I see that some mention has been made of earlier projects (e.g. COMESA). However, I suggest looking

back a bit to the issue of supersonic transport (which is indeed discussed again today, see some of the citations in this paper). But in the seventies a controversy about supersonic transport had started in the United States. At that time, large fleets of stratospheric supersonic aircraft were planned (US: Boeing, Britain/France: Concorde, Soviet Union: Tupolev) and a fleet of 500 supersonic planes seemed a reasonable estimate. It is interesting to note that the concern was an enhanced catalytic ozone destruction; originally ozone destruction by OH and $HO_2$ radicals (resulting from the release of water vapour in the engine exhausts, like discussed in this manuscript for hypersonic transport) was considered, but it was soon realised that the catalytic destruction of ozone by $NO_x$ posed a much greater threat to the ozone layer (Johnston, 1971; Crutzen, 1972). Indeed this issue was part of the motivation of Crutzen (1970) to investigate the impact of $NO_x$ on the ozone layer. Perhaps some effort to touch upon this history might be helpful to the paper.

**Some details**

- l. 3: it would be helpful to give approximate numbers for these emission.

- l. 6: if 15 km is in the tropics, months seems rater long, on the other hand months is short for emissions at (say) 30 km in the stratosphere. Perhaps one could be a bit more specific here.

- l. 8: I would not include (potential) speculations in the abstract – concentrate on the new findings of the paper.

- l. 350: what about the loss of $H_2O$ in the Antarctic stratosphere in winter (e.g., Kelly et al., 1989; Poshyvailo et al., 2018) could this loss process be of relevance for the considerations here? Is it implemented in the models?

- l. 356: Here you say that higher altitudes have a negligible effect on the mass perturbation, but in l 361 you say that the "$H_2O$ mass perturbation is approximately twice as large for the higher flying aircraft compared to the lower flying aircraft..." – isn't this a contradiction? I think this could be better explained.

- l. 370, Fig 6: here and elsewhere: $H_2O$ should not be in italics in the figures.

- l. 424, Table 4: ozone in percent; do you mean total ozone here?

- l. 522, Fig. 12: The figure shows an enhancement factor, which is not explained in the caption. No unit is given. However, a little below (l. 532) a unit is given in the text, and the values are compared to Fig. 12: I find this hard to follow.

- l. 575: This is not the most recent edition of this book; see Brasseur and Solomon (2005).

- l. 617: How is this reference available?

- l. 634/637: Journal missing?

**References**

Archer, D. and Brovkin, V.: The millennial atmospheric lifetime of anthropogenic $CO_2$, Clim. Change, 90, 283–297, 2008.

Brasseur, G. and Solomon, S.: Aeronomy of the Middle Atmosphere: Chemistry and Physics of the Stratosphere and Mesosphere, Springer, Heidelberg, Germany, third edn., 2005.

Crutzen, P. J.: The influence of nitrogen oxides on the atmospheric ozone content, Q. J. R. Meteorol. Soc., 96, 320–325, 1970.

Crutzen, P. J.: SSTs–a threat to the earth's ozone shield, Ambio, 1, 41–51, 1972.

Johnston, H.: Reduction of stratospheric ozone by nitrogen oxide catalysts from supersonic transport exhaust, Science, 173, 517–522, 1971.

Kärcher, B.: Aircraft-generated aerosols and visible contrails, Geophys. Res. Lett., 23, 1933–1936, 1996.

Kelly, K. K., Tuck, A. F., Murphy, D. M., Proffitt, M. H., Fahey, D. W., Jones, R. L., McKenna, D. S., Loewenstein, M., Podolske, J. R., Strahan,

S. E., Ferry and K. R. Chan and J. F. Vedder, G. V., Gregory, G. L., Hypes, W. D., McCormick, M. P., Browell, E. V., and Heidt, L. E.: Dehydration in the lower Antarctic stratosphere during late winter and early spring, 1987, J. Geophys. Res., 94, 11 317–11 357, 1989.

LeTexier, H., Solomon, S., and Garcia, R. R.: The role of molecular hydrogen and methane oxidation in the water vapour budget of the stratosphere, Q. J. R. Meteorol. Soc., 114, 281 – 295, 1988.

Poshyvailo, L., Müller, R., Konopka, P., Günther, G., Riese, M., Podglajen, A., and Ploeger, F.: Sensitivities of modelled water vapour in the lower stratosphere: temperature uncertainty, effects of horizontal transport and small-scale mixing, Atmos. Chem. Phys., 18, 8505–8527, https://doi.org/10.5194/acp-18-8505-2018, 2018.

Riese, M., Ploeger, F., Rap, A., Vogel, B., Konopka, P., Dameris, M., and Forster, P.: Impact of uncertainties in atmospheric mixing on simulated UTLS composition and related radiative effects, J. Geophys. Res., 117, D16305, https://doi.org/10.1029/2012JD017751, 2012.

Solomon, S., Rosenlof, K., Portmann, R., Daniel, J., Davis, S., Sanford, T., and Plattner, G.-K.: Contributions of stratospheric water vapor to decadal changes in the rate of global warming, Science, 327, 1219–1223, https://doi.org/10.1126/science.1182488, 2010.

---

## Author Response (AR1)

**Authors Response**

Dear Reviewer,

we would like to thank you very much for your helpful comments and we included your comments in the revised version. The answers are given in *blue and italic* below each comment.

Yours sincerely,

Johannes Pletzer

**Review of "The Climate Impact of Hypersonic Transport"**
**by Pletzer et al.**

**General Comments**

The paper by Pletzer et al. provides a modeling study of the potential climate and ozone impacts due to emissions from Hypersonic Transport. The study is motivated by the growing contribution from aviation to climate change and also increasing desire to reduce air travel time. The present paper evaluated two hypersonic aircraft designs using two 3D chemistry-climate models, providing an assessment on climate and ozone impact for flying these aircraft. The present paper shows for the first time that a net chemical production at high altitudes leading to an increase of H2O perturbation lifetime and H2O concentration.

This is a very interesting and well-written paper that is a significant contribution that merits prompt publication. I only have a few minor suggestions for the authors to consider (below). What I find 'missing' is some contextual information. Also, I suggest that the limitations of the models need to be discussed more explicitly. I have detailed some of this below that might easily rectify this for the authors to consider.

**Specific Comments**

In the methods and simulations section, it is still not clear to me what nudging is applied in the models? I am assuming U wind, V wind and surface pressure? Is there anything else? Is it nudged all the way from surface to the model top? I assume the ERA-Interim meteorology fields first get interpolated to model grid (both horizontally and vertically) offline before nudged to the models? Please explain/clarify in the paper.

*Thank you very much for this very positive feedback. We extended the information on the nudging setup with the following text:*

*"EMAC is based on the spectral transform dynamical core of ECHAM. The nudging is applied by Newtonian relaxation of the prognostic variables divergence, vorticity, temperature and the logarithm of the surface pressure in the spectral representation (spherical harmonics) of these variables towards the ECMWF ERA-5 reanalysis data . \textit{u} and \textit{v} wind are derived variables calculated through derivation and spectral transformation. The wave-0 of temperature (global mean) is omitted, there is further no nudging applied on the sub-synoptic scale, aiming at an optimal compromise between observed (i.e. reanalysed) and simulated meteorology.*

*Moreover, the nudging is applied in vertical direction only between the 4th model layer above the ground and approximately 200 hPa, in order to avoid inconsistencies in the planetary boundary layer and to let the stratosphere develop freely and driven by the tropospheric wave activity. This nudging setup is identical to that of the RC1SD-base-10 simulation described by \citet{Joeckel2016}. The ECMWF ERA-5 reanalysis data are preprocessed by spectral transformation and truncation for the applied model resolution."*

*The information that, first, only u and v wind components are nudged from 1000 hPa to 1 hPa in LMDZ-INCA was added to the text and, second, that the winds are interpolated on the horizontal grid in preparation of the run and are interpolated on the model pressure levels at each time steps.*

L151
"The respective relaxation times are listed in 1."
The respective relaxation times are listed in Table 1?

*The missing 'Table' was added.*

L190 and L238
Two models seem to have large difference in derived lightning NOx – 0.2 TgN/yr and 5.5 TgN/yr in EMAC and LMDZ-INCA respectively. Why is that? How do these two models derived the lightning NOx. How does this large difference can potentially influence the results here, if any? Hope the authors could elaborate more on this.

*We added further information on the parametrizations in the text. The Grewe parametrization in EMAC is based on convective mass-flux and in LMDZ-INCA on convective cloud heights. We added a comparison in the 'Simulations'-subsection. However, this is not where the difference comes from. The difference was actually an error in the selection of the correct variable. We corrected the lnox value from EMAC, which is 5.0 Tg and not 0.2 Tg. 5.0 Tg also agrees with Schumann and Huntrieser, who estimate the annual average lightning NOx production to be between 5 and 11 TgN/yr. Values of both models agree well with the range of the recent CMIP6 models: 3.2-7.6 TgN [Ref. IPCC AR6 chapter 6 (section 6.2.2.1)].*

L194
What are the vertical resolutions (especially in the upper troposphere and stratosphere) in the two models? What is the vertical resolution of the HIKARI aircraft emission inventory? What is the underline assumption of the aircraft emission spread throughout the model grid? What is the role of model diffusivity in the distribution of emissions? What is the limitation? More discussion on this aspect would be helpful.

*The vertical resolution of EMAC in the applied T42L90MA resolution is approximately 550 m in the UTLS region, reaches 1200 m at the stratopause and increases to 3200 m in the mesosphere.*

*The vertical resolution of LMDZ-INCA in the applied resolution is approximately 1000-1300 m in the UTLS region, reaches 5000 m at the stratopause and increases to 8700 m in the mesosphere*

*We added information on the vertical resolution of both models and the horizontal and vertical resolution of HIKARI data.*

*Due to the rather coarse model resolutions of CCMs and CTMs (a sacrifice for the computationally expensive chemistry calculations) there is no horizontal diffusion, neither explicit nor parameterized, since the numerical diffusion of the applied large scale advection algorithm cannot be avoided. Vertical diffusion is parameterized for the vertical mixing in the planetary boundary layer.*

L240 to L250
The model uses the meteorology data from the time period 2000-2014 to simulate climate impact in 2050-2064. One can argue that using the meteorology fields between 2000 to 2014 doesn't account for the dynamics changes (e.g., temperature and circulation) at the upper troposphere and lower stratosphere owning to climate change. For example, previous studies have shown that the Brewer–Dobson circulation has been strengthening in response to climate change (e.g., Butchart et al., 2006; Shepherd et al., 2011). How much difference can it make to the resulting effects?

Butchart, N., Scaife, A. A., Bourqui, M., De Grandpré, J., Hare, S. H. E., Kettleborough, J., ... & Sigmond, M. (2006). Simulations of anthropogenic change in the strength of the Brewer–Dobson circulation. *Climate Dynamics*, 27(7), 727-741. DOI 10.1007/s00382-006-0162-4

Shepherd, T. G., & McLandress, C. (2011). A robust mechanism for strengthening of the Brewer–Dobson circulation in response to climate change: Critical-layer control of subtropical wave breaking. *Journal of the Atmospheric Sciences*, 68(4), 784-797. https://doi.org/10.1175/2010JAS3608.1

*Thank you for this comment. We added another subsubsection in the new discussion subsection 'Limitations of the Model Simulations'.*

*\subsubsection{Strengthening of the Brewer-Dobson Circulation}*

*In our model simulations we use atmospheric composition projections for the years 2050-2064 combined with present day meteorology (2000-2014). Hence, projections of the dynamic component are not included in our simulations. The main reason is, that reanalysis data from the future is simply not available for nudging. Using another method was not an option, since we rely on nudging to have the same meteorology in both models for a high signal-to-noise ratio. However, the changes of dynamic processes like the Brewer-Dobson circulation due to climate change are very likely significant and we therefore discuss the topic briefly \citep{butchart_simulations_2006,shepherd_robust_2011}. The associated transport is the dominant factor of water vapour perturbation lifetime and therefore of the climate impact of hypersonic aircraft. An increase in strength of the stratospheric and mesospheric circulation would most likely reduce the climate impact of hypersonic aircraft. \citet{butchart_simulations_2006} estimate the troposphere-stratosphere mean mass exchange rate to increase with 2 \% per decade (with considerabe differences between the models). That would result in an approximately 8-10 \% stronger circulation from 2050-2064 and in turn -- if the effect can really be directly translated to perturbation lifetime -- the climate impact of hypersonic aircraft would be reduced by approximately the same percentage, i.e. the water vapour perturbation lifetime and associated*

*RF for the LAPCAT version calculated by EMAC might be reduced from 4.6 years and 40 mW/m2 to roughly 4.2 years and 36 mW/m2 when considering an enhanced Brewer-Dobson circulation.*
*.*

Note: you should also add a few sentences about why you omitted the dynamical change (you had to!) because there are no "nudging" data available for the future, and the nudging was important for the signal-to-noise ratio … and it had to be "the same meteorology" in both cases for this … (thus, even if we would have nudged to dynamical data from a future EMAC simulation, eg. RCP-8.5 or similar, this would not have helped, since then the meteorological sequence is different and the signal-to-noise ratio goes down ...

Figure 1 shows the vertical sum of H2O emissions over all vertical levels. How different the figure would be if one only plotted the emissions at cruise altitudes?

*To our knowledge, potential hypersonic flight routes were identified in the HIKARI project. However, the city-pairs were not used in the 3 dimensional HIKARI data. The figure limited to e.g. stratospheric altitudes would therefore show very similar features like the vertical sum over all levels, excluding take-off and landing emission features in close proximity to the cities. We confirmed this with a world map plot, where only the stratospheric levels were integrated.*

*LAPCAT Emission Data*

[Figure]

[Figure]

*Integration over all vertical levels*         *Integration over selected vertical levels*

*We added the following sentence in the HIKARI section as additional information: 'All aircraft designs travel from and to the same city pairs'*

L295
States that "Most of the emitted trace gases are transported to tropospheric altitudes". I am curious how much emitted H2O is chemically destroyed in the stratosphere? Have the authors explored the ratio of emitted stratospheric trace gases transported to tropospheric altitudes VS chemical destruction in the stratosphere? Which is the major contributor (transport or chemistry) to the stratospheric H2O sink here? It would be very interesting to see.

*The major sink is transport to the troposphere. The chemical loss of water vapour is overcompensated due to the $H_2O$ recombination and enhanced methane depletion (see table). Parts of the sentences might be misleading and are changed accordingly. Thank you for*

*pointing that out.*

|            | Emission>18 km | Prod. | Loss   | Loss to Troposphere |
|------------|----------------|-------|--------|---------------------|
| LAPCAT     | 13.7           | 4.77  | -4.01  | -14.46              |
| ZEHST      | 21.2           | 14.64 | -12.69 | -23.15              |

*'Most of the emitted trace gases are transported to tropospheric altitudes and only parts of the annual perturbation remain.' → 'During the fifteen years, the emitted trace gases, while being chemically converted, are continuously transported to tropospheric altitudes and only parts of the total emitted trace gases remain as changes to atmospheric composition.'*

Table 3
It would be interesting to see how much percentage of total H2O perturbation stay in the atmosphere (above the tropopause) when it reaches equilibrium (e.g., Perturb. LAPCAT/ total LAPCAT emission *100% in two models)

*We calculated how much of the annual H2O emission stays above the tropopause using the perturbation lifetime and the annual perturbation from Table 3. The ratio is listed in the following table.*

| Scenario | EMAC    | LMDZ-INCA |
|----------|---------|-----------|
| ZEHST    | 71.4 %  | 64.3 %    |
| LAPCAT   | 78.3 %  | 76.2 %    |

Table 4
shows that LMDZ-INCA calculates a higher ozone destruction than EMAC. The difference can be about 42% for lower flying aircraft (ZEHST). What factors can be mainly contributed to the difference derived in two models? Could the authors add a few sentences to discuss about this?

*The main reason is the fact that EMAC produces an increase of ozone in the lower stratosphere. In terms of mass this is significant fraction of the total. Therefore the total loss is higher in LMDZ-INCA. We have to add that the caption was not up to date, since we present the relative total ozone changes, not the values above the tropopause, which we corrected.*

L476
"over four year"
over 5 years? 2010-2014

*Changed to 'over five years'.*

A recent study by Zhang et al., 2021 calculated the stratospheric adjusted radiative forcing of water vapour and ozone perturbation at difference cruise altitudes. How does the water vapor and ozone radiative forcing impact in this study compared to their study? Scaling by per unit H2O emission in $mW(m^2Tg)^{-1}$?

Zhang, J., Wuebbles, D., Kinnison, D., & Baughcum, S. L. (2021). Stratospheric ozone and climate forcing sensitivity to cruise altitudes for fleets of potential supersonic transport aircraft. *Journal of Geophysical Research: Atmospheres*, 126(16), e2021JD034971. https://doi.org/10.1029/2021JD034971

*To compare the results we used the pieces of information given in Zhang et al, which are fuel use, the $H_2O$ emission index and radiative forcing for 19-21 and 21-23 km to calculate the linear relation and extrapolate this to cruise altitudes of the aircraft ZEHST and LAPCAT:*

*EI ($H_2O$) = 1,237 g $H_2O$/kg fuel burn*
*→ 47.18 Tg Fuel → 47.18 Tg Fuel \* 1.237 Tg $H_2O$ / Tg Fuel = 58.36 Tg $H_2O$*

*Radiative forcing*
*At 20 (19-21) km: 1.00 mWm$^{-2}$ / Tg Fuel → 47.18 mW m$^{-2}$*
*At 22 (21-23) km: 1.12 mWm$^{-2}$ / Tg Fuel → 53.1 mW m$^{-2}$*

*Linear Trend:*

*At 20 (19-21) km: 47.18 mW m$^{-2}$ / 58.36 Tg $H_2O$ → 0.81 mW m$^{-2}$ / Tg $H_2O$*
*At 22 (21-23) km: → 53.1 mW m$^{-2}$ / 58.36 Tg $H_2O$ → 0.91 mW m$^{-2}$ / Tg $H_2O$*

*Delta RF / delta km = (0.91 – 0.81) mW m$^{-2}$ / Tg $H_2O$/ (23 – 21) km = 0.05 mW m$^{-2}$/ Tg $H_2O$ / km*

*Extrapolation to 26 and 35 km:*

*At 26 km: 0.91 mW m$^{-2}$ / Tg $H_2O$ + 0.05 mW m$^{-2}$ / km \* (26 – 22) km = 0.93 mW m$^{-2}$*
*At 35 km: 0.91 mW m$^{-2}$ / Tg $H_2O$ + 0.05 mW m$^{-2}$ / km \* (35 – 22) km = 1.56 mW m$^{-2}$*

*We added a subsection on this topic ($H_2O$ only) in the discussion.*

*\subsection{Comparison to Emission at Lower Altitudes}*

*\citet[Figure 11]{Zhang2021} published an altitude dependent comparison of ozone and water vapour RF normalized to fuel use. To compare their results on climate impact we used their emission index (EI(H$_2$O) = 1,237 g(H$_2$O)/kg fuel) and fuel use (47.18 Tg) to recalculate their results to RF per emitted water vapour in teragram. With the above values, we extracted the linear relation of 0.1 \unit{mW\,m^{-2} / 2 km} for an increase of RF with altitude. The extrapolation to ZEHST and LAPCAT cruise altitudes resulted in 0.93 \unit{mW\,m^{-2}} and 1.56*

*\unit{mW\,m^{-2}} for ZEHST and LAPCAT, respectively. Compared to our results (1.1-1.5 and 1.6-1.9 for ZEHST and LAPCAT, respectively), presented in Table \ref{table_rf}, the here calculated values are generally lower than EMAC results, especially compared to ZEHST. LMDZ-INCA results are lower for ZEHST compared to the extrapolation, however agree astonishingly well for LAPCAT. Clearly, the linear relation is not a perfect fit, however it shows the same trend and the order of magnitude agrees very well.*

L563
"To conclude briefly, the impact on climate of aircraft emitting water vapour and flying above the tropopause increases very much with altitude." Agree with the authors: this conclusion is indeed valid for climate impact of water vapour emission with altitude; however, it might not be the case for ozone perturbation. Zhang et al study showed that there is an inflection point where the radiative forcing of ozone perturbation changes from positive to negative with increasing the flying altitude.

*This is indeed true and an inversion point has been shown before in idealized simulations (not aircraft emission scenarios) by Lacis et al in "Radiative forcing of climate by changes in the vertical distribution of ozone". To clarify this topic, we extended the sentence.*

'To conclude briefly, the impact on climate of aircraft emitting water vapour and flying above the tropopause increases very much with altitude, since water vapour radiative forcing is significantly larger than ozone radiative forcing, which has a more complex altitude dependency, and methane radiative forcing.'

In Kinnison et al. 2020 study, they have showed in figure 1 that the natural stratospheric photochemical production of water vapor resulting from the oxidation of methane is 60 Tg (H2O)/year, which seems to be much higher than the value (less than 14.6 Tg/yr) presented in Figure 7 and Figure 8 in this study. Could the authors explain a little why there is such a big difference?

*60 Tg(H2O)/year should be the natural production in the model, shouldn't it? The values presented in Figure 7 and Figure 8 are the changes to the model's natural production (scenario-reference) and are therefore considerably smaller. To clarify, we added '(scenario-reference)' in both Figures.*

The simulations assume an atmosphere without volcanic eruptions, but volcanic eruptions will happen in the time frame of an employment of hypersonic transport. What are possible effects of aerosol particles and water vapour of volcanic origin? How would this effect change the picture put forward here? Perhaps nothing can be said, but then this point should be clearer as a limitation of the study.

*We added the following subsubsection in the discussion subsection 'Limitations of the Model Simulations':*

\subsubsection{Aerosol and Water Vapour from Volcanic Origin}

In our model simulations, we assume an atmosphere without volcanic eruptions. However, volcanic eruptions could occur during the decades of operation of new aircraft. Volcanic emissions, like water vapour or sulphate aerosols, affect the atmospheric composition in the stratosphere, especially through heterogeneous chemistry, and these changes are strongly dependent on latitude and season. The changes of lower stratospheric water vapour changes due to volcanic eruptions is on the order of two years and affect ozone concentrations \citep{Stenke2005}. Sulfate aerosols are known to increase temperatures in the tropics and could in turn enhance the Brewer-Dobson circulation, eventually slightly reducing the climate impact of hypersonic transport. Overall, the topic is very complex in itself and how hypersonic emissions and volcanic emissions influence each other remains to be answered with robust and topic specific simulations.

L800
"over a time period 800 of fourteen years (2000-2014)"
over a time period 800 of fifteen years (2000-2014).

*Added 'fifteen'.*

L812
https://www.iagos.org/partners
The link doesn't work for me

*Added a href with updated link '\href{https://www.iagos.org/organisation/members/}{IAGOS member institutions}'*

**Authors Response**

Dear Reviewer,

we would like to thank you very much for your additional comments. Our answers are given in *blue and italic* below each comment.

Yours sincerely,

Johannes Pletzer

**Review of "The Climate Impact of Hypersonic Transport"**
**by Pletzer et al.**

authors have presentd a comprehensive study about the climatic impacts of air pollution from hypersonic aircrafts. Authors have covereddetailed explanation about the two set of emissions, two numerical models and the climatic impact. The reviewer 1 made very specifc and accurate observations and I agree that this manuscript may needs minor modification.

*Thank you very much for your positive feedback.*

In particular:

On line 131, authors states: "A comparison with satellite data shows that over the annual cycle ozone volume mixing ratios are well reproduced in the stratosphere, apart from southern polar regions and with larger differences at tropospheric altitud". Can you comment about these differences?

1. *The stratospheric southern polar bias is larger in free running EMAC simulations and especially low in simulations with specified dynamics (without mean temperature nudging). Hence, our chosen EMAC model setup is especially well suited for our application of modelling stratospheric ozone.*
2. *The troposphere bias is reduced in simulations with specified dynamics, where mean temperature nudging is included. According to Joeckel et al (2016), this originates from the complex relation of reduced vertical convective activity, shift in tropopause height, reduced lightning NOx and eventually increased NOx emission from soil.*

*The whole topic is described extensively in Jöckel et al (2016), sections 4.7 and 5. We added the additional information on bullet point 1 in the publication.*

Include color legend for figure 1 and 2.

*The reason, why the colormap was omitted, is stated in the figure caption.*

Suggestion: It is better to communicate the results with bar plots and not pie charts.

*Generally, we agree. However, in this specific case, we decided to use a pie chart, first, simply not to use bar plots twice and, second, the two absolute values of the pie chart (4.7 Tg, 14.6 Tg) do not have to be normalized to be able to compare the reactions ratio with one look.*

Lines 419-421. Is EMAC better? I believe the author plays with the ideia without explicitly

stating.

*Both models are well validated within the Coupled Model Intercomparison Project (CMIP). It is a huge advantage to be able to use two climate models. With results from two models, we can compare the results, validate trends and reduce the error of our estimates. We do not agree and do not want to transport the view that EMAC or LMDZ-INCA is better. If you do have a proposition on how to change the text, we would gladly welcome it, since it is not our intention to rank the models. Nonetheless, we rephrased the passage, hoping to address the issue.*

*Old: "To conclude, in the EMAC results, either the uncertainties due to annual variability is larger compared to LMDZ-INCA for 420 both O3 and NOx or the perturbation to background ratio is larger. Additionally, the larger vertical resolution in EMAC could explain the different uncertainty value (p < 0.05 for EMAC, p < 0.001 for LMDZ-INCA)"*

*New: "To conclude, the uncertainties due to the annual variability are lower in LMDZ-INCA compared to EMAC for both \chem{O_3} and \chem{NO_x}, highlighting the significance of LMDZ-INCA results. The larger vertical resolution in EMAC could explain the different uncertainty value (p < 0.05 for EMAC, p < 0.001 for LMDZ-INCA), since it allows more detailed perturbation patterns, which is not model-specific, but resolution-specific."*

Line 486: The author mention WACCM used by another author without define it.

*We define the model (Line 474), but do not compare the specific model setup to our model setups. We added additional information for the readers (underlined text).*

*"[…] using the coupled chemistry-climate model WACCM (Whole Atmosphere Community Climate Model). Similar to our model setups, WACCM simulates atmospheric chemistry and dynamics. For more detailed information, please refer to their publication."*

There are some paragraphs consisting in less than 2 phrases. Each paragrah should have at least thee phrases, intro, body and conclusion.

*Thank you very much for the suggestion. We have changed the composition of the paragraphs to reduce the occurrence of very short text passages.*

The manucrsipt has an excessive number of tables and figures. Consider moving some into supplementary material

*We are strongly convinced that especially figures are helpful in understanding and comparing the main text. Therefore, we prefer to keep the figures and moved Table 7 (RF including upper tropospheric perturbation) to the appendix.*

**Authors Response**

Dear Reviewer,

we would like to thank you very much for your analysis of our manuscript and broad understanding of the topic. Our answers are given in *blue and italic.*

Yours sincerely,

Johannes Pletzer

**Review of "The Climate Impact of Hypersonic Transport" by Johannes Pletzer et al.**

**General remarks**

This paper analyses the Climate Impact of Hypersonic Transport. Very few studies in this direction are available. Therefore such studies are needed and such results certainly constitute an important contribution to ACP. It is also good that two independent models are employed (and the results compared) as specific model features could impact the results of such model studies.
However, as the paper stands, I do not think it is very useful. My most important criticism is that the message (result) of the paper is not clear (see also below). Some issues (e.g. contrail formation by subsonic planes) seem to be overlooked and it is not clear to the reader how the "impact" is quantified and measured. I am also not convinced about the "first time that a recombination to H2O" is found (see equation 1), but such issues should be clearly worked out, not just mentioned in passing. (I note that the key reaction, equation 1, if I understood correctly) is not even explicitly mentioned in the manuscript.

*We are thankful that the reviewer acknowledges the need of such studies and highlights the importance of the use of two independent models.*
*Message of the paper: This is an important point. It was not clear to us that our message was not conveyed correctly. The detailed changes to address this topic can be found below.*

*Contrail formation: We added another paragraph in the discussion subsection 7.1. The background is the following. A study by Stenke et al (2008) estimates the change in contrail formation and the change of contrail radiative forcing for subsonic and supersonic aircraft. According to the authors the change in contrail radiative forcing and total contrail cover is very small. They report a shift of contrail cover from mid latitudes subsonic cruise levels to low latitudes supersonic cruise levels (see enclosed Figure from that paper). For hypersonic aircraft this relation might be changed since those aircraft are flying above the tropical tropopause and hence do not form contrails in the tropics. Therefore, the replacement would probably lead to a reduction of contrail radiative forcing. See Figure 3 from Stenke et al (2008) showing the total contrail cover in percent for supersonic aircraft, i.e. S5 (right):*

[Figure]

**Fig. 3.** Annually averaged total contrail cover [%] for the subsonic fleet S4 as simulated by ECHAM4 (left), and differences in total contrail cover [%] between S5 and S4 (right). The displayed differences are significant at the 95% level (t-test).

*Very importantly, it seems that a misunderstanding exists: We certainly did not intend to claim to have found a new reaction for the recombination to $H_2O$. Instead we want to convey the overcompensation mechanism, counteracting to the expected photochemical $H_2O$ loss as indicated by Brasseur and Solomon (production-loss). Hence the discussion of the rate of those reactions is the new finding that determines the perturbation loss time and not the reaction itself.*

I am sorry for sounding so negative; I think a lot of work on the manuscript is required, but the topic is of great importance.

**Comments**

Message of this paper
It is not clear to me what the main message of the paper is. To me it seems that the question is the radiative forcing induced by a fleet of hypersonic planes because of the emissions of $H_2O$ they cause in the stratosphere. A lot of the discussion is along this argument. But then the abstract talks about depletion of the ozone layer (l. 14, but is this depletion in column ozone?) without addressing the processes (is the depletion caused by $NO_x$ or by $HO_x$ or both)? Is the depletion relevant because of UV issues (the ozone reductions seem small) or because of radiative forcing?

Thank you very much for this helpful comment. We revised the abstract accordingly to focus explicitly on the radiative forcing change.

"We find a 18.2 $\pm$ 2.8 mW m$^{-2}$ and 36.9 $\pm$ 3.41 mW m$^{-2}$ increase in stratosphere adjusted radiative forcing due to the two hypersonic fleets flying at 26 km and 35 km respectively. Ozone changes contribute 8.0-21.8 % and water vapour changes contribute 78.3-92.0 % to the warming on average."

The impact of water vapour and NOx emission on ozone has been extensively described by and focused on by Kinnison et al considering also their impact independently from each other. They showed that the NOx emission has a considerably larger impact on ozone than water vapour emission at altitudes from 30-40 km. In our publication, we focus on the radiative forcing changes, while addressing the similarities and differences between our model simulations and Kinnison et al. (e.g. subsection 7.1 "Atmospheric Composition Changes" and Fig. 6 "Photochemical water vapour lifetime"). The relevance of NOx or HOx in the depletion of ozone is indeed important and hence we dedicate another publication to this topic, focusing on altitude and latitude sensitivities.

According to the results from McKenzie et al (1991) one percent of ozone depletion cause approximately

1.26-1.40 % increase in erythemal irradiance (EI). For the average total ozone depletion this is equal to an increase of 0.10-0.12 % in EI for the ZEHST aircraft and to 0.20-0.22 % EI for LAPCAT.

[Figure]

Fig. 4. From McKenzie et al (1991)

The climate impact is measured relative to subsonic aircraft (l. 16/17), but what quantity is used to calculate the relative impact? Is it radiative forcing? This should be clear from the abstract. Assuming it is radiative forcing does this forcing only consider the impact of $CO_2$ emissions by conventional subsonic aircraft? Such aircraft cause contrails (ice particles) which have a potential impact on radiative forcing (e.g., Kˊarcher, 1996) – has this effect been considered?

Thank you for pointing that out. We added the pieces of information "stratosphere adjusted radiative forcing" and "mean-surface temperature change" to the abstract and the introduction.

The relative impact is calculated using AirClim. The climate impact of subsonic aircraft includes contrail formation, CO2, H2O and NOx effects (short-lived ozone, primary mode ozone, methane). Further information is available from Grewe & Stenke (2008) and Dahlmann et al (2016). Note that also the efficacies of the individual species are taken into account as described by Dahlmann et al (2016) that are close to those of Lee et al (2021).

Further, there could be ways to manufacture carbon neutral kerosene like fuels (for subsonic aircraft as well as for supersonic and hypersonic aircraft) and of course a subsonic aircraft in the future which is fuelled by (green) liquid hydrogen is not unthinkable. I understand that this study cannot discuss all of these possibilities, but by making a particular choice, it rans into the danger of giving a biased comparison. And further (see below and the discussion in the manuscript) the lifetimes of radiative forcing of emissions of $H_2O$ and $CO_2$ to the atmosphere are different (and depend very much on altitude in case of $H_2O$).

Clearly, your comment to use carbon neutral kerosene like fuels for high-flying aircraft is valid. We decided to focus on hydrogen powered aircraft for two reasons. First, current hypersonic transport projects focus, to our knowledge, on cryogenic propulsion. Second, kerosene fuel would reduce the emission of water vapour by a factor of three and would require a new design of propulsion technology and new aircraft design (larger take-off weight). These topics are being addressed in the EU-project MORE&LESS. Water vapour would most probably remain the main climate driver or one of the main climate drivers besides ozone and can be estimated with our radiative forcing sensitivity RF/Tg(H2O) Emission).

A comparison of subsonic aircraft fueled with sustainable aviation fuel and hypersonic aircraft would result in a significantly larger ratio. First, because they would be both carbon neutral and, second, because the contrail radiative forcing is likely being reduced since sustainable aviation fuel (SAF) leads to lower particle number densities relating directly to fewer ice crystal number densities in the contrail and implying shorter lifetimes and smaller radiative effects.

Still it is a valid point and we changed the title of the publication to "The Climate Impact of Hydrogen Powered Hypersonic Transport" to avoid misunderstandings.

**Carbon cycle**

It is stated in the paper that "the $CO_2$ perturbation originating from fossil fuel is subject to a large variety of sinks with different lifetimes. In general, the range is approximated with 2-20 centuries, where most of the $CO_2$ climate impact is taken up by ocean and biosphere sinks and 20-35 % remain in the atmosphere for longer time . . . "

First, it should be noted that the $CO_2$ that remains in the atmosphere can only be really taken out of the system by sedimentation of carbon containing material to the ocean sediments on timescales much longer than centuries (100 000 years) and, second, the ocean uptake depends on the ocean circulation and ocean water chemistry (which is in the order of perhaps 5000 years) (Archer and Brovkin, 2008).

*Your comment is very much appreciated. We already wanted to oppose the often-transported message that $CO_2$ has a lifetime of approximately 80-120 years, which is not true. That is why we addressed the long tail of $CO_2$ lifetime. Our description was vague with "for longer time". We happily extended the description with another sentence ("Hence, released $CO_2$ will affect climate for tens of thousands to hundreds of thousands of years").*

**Water vapour as a greenhouse gas**

I think we all agree that water vapour is the most important greenhouse gas; it accounts for about half of the present day greenhouse effect and is the most important gaseous source of infrared opacity in the atmosphere. Tropospheric water vapour warms the climate and so does lower stratospheric water vapour (Solomon et al., 2010; Riese et al., 2012). However, it seems to be that an underlying assumption throughout the paper that stratospheric $H_2O$ always warms. However, at higher stratospheric altitudes and especially in the tropics a unit mass increase in $H_2O$ cools the climate (Riese et al., 2012). The authors might not agree with the results by Riese et al. (2012) but this aspect of heating and cooling should be discussed in the paper.

*Thank you very much for the reference by Riese et al. Since the main mass perturbation of $H_2O$ emission is accumulating in the lower stratosphere and only small mass perturbations appear in tropical or higher stratospheric regions (Fig. A2, Fig. A3), we, as a matter of fact, focus on the warming. We gladly agree to include the information stated by Riese et al to convey a complete picture to the readers. In our opinion the main picture remains the same. We additionally referenced the Figures on the mass change in the main text, which was missing.*

*We extended the text in the section "Radiation and Climate" with an introduction, which relates atmospheric composition to radiative forcing.*

*"The radiative forcing caused by atmospheric composition changes depends very much on location (Lacis et al, 1990; Riese et al, 2012). In our results, the largest water vapour concentration change appears at lower stratospheric altitudes and increases poleward (Fig. A2, Fig. A3). The differences between the ZEHST and LAPCAT scenario, except for magnitude, are small. Additionally, we describe the ozone increase in the lower tropical stratosphere, where the RF sensitivity and air density are larger (concentration changes not shown). Hence, according to (Lacis et al, 1990; Riese et al, 2012) we expect a warming for both ozone and water vapour changes."*

Overall, I suggest that the background balance of water vapour through-out the stratosphere is considered (e.g., LeTexier et al., 1988; Brasseur and Solomon, 2005; Poshyvailo et al., 2018, and references therein), on top of which the impact of perturbations by the proposed fleet of hypersonic planes can be assessed.

The water vapour budget according to LeTexier et al (1988) has been re-evaluated with three different approaches using EMAC by Frank et al (2018). This includes simulations with full chemistry very similar to ours, which confirm the findings by LeTexier et al. The methane oxidation submodel that we applied in our setup (CH4), was developed by Winterstein & Jöckel (former Frank).

For completeness, we address the background balance of water vapour in the revised introduction (paragraph line 50).

"The middle atmospheric balance of water vapour is determined by methane oxidation, photochemical lifetimes of HOx compounds and tropical upward transport, which is limited by the coldpoint temperature (LeTexier et al, 1998; Brasseur, 2005; Frank et al, 2018). Polar dehydration by polar stratospheric clouds and the sedimentation of the particles contribute to the balance."

In our opinion the UTLS region is a special case, since large uncertainties remain that projects like TPCHANGE are trying to address. We used the model evaluation with IAGOS to assess our models performance in the extratropical UTLS. Further we addressed the variability of upper tropospheric water vapour and included an upper estimate of radiative forcing based on the variability (Tab. 7).

**Water vapour production in the stratosphere**

The authors state in the abstract that "$H_2O$ depletion at high altitudes is overcompensated by a recombination of hydroxyl radicals to $H_2O$ . . . " and later in the paper state: "Opposite to the expected removal of $H_2O$ emissions, we found a before unknown net-recombination of $H_2O$" (l. 391). I think that they are referring to the reaction

$$HO_2 + OH \dashrightarrow H_2O + O_2 \quad (1)$$

First, given that this seems to be a major issue for the paper I suggest explicitly stating the reaction. Second, the water vapour production in eq. 1 is known (see, e.g., eq. 5.105 in Brasseur and Solomon, 2005). So I am not sure what "before unknown" means here. Finally, I agree with the authors that the lifetime of water vapour decreases with altitude, but of course water vapour concentrations are a balance of loss and production terms (like it is the case for other species), so I am also not sure about "expected removal of $H_2O$ emissions".

- *We added the reaction right before Fig. 7.*
- *The water vapour production is known in literature and has of course been implemented in both models, EMAC and LMDZ-INCA for a long time. Thank you for pointing out that we did not state the reaction explicitly. We want to convey that the overall net-recombination as a follow up to hypersonic emission was not known before. We highlight three processes, increased methane oxidation, increased nitric acid oxidation, and HOx recombination to explain the overcompensation. Expectations for the climate impact in regard to the photochemical destruction of water vapour were published before (Steelant et al, 2015). Here, we want to address researchers with aircraft design background in addition to researchers on atmosphere with a broader knowledge of atmospheric processes.*

*Former: „* Opposite to the expected removal of H2O emissions, we found a before unknown net-recombination of H2O. Both models show an increase in H2O perturbation lifetime and H2O perturbation at the higher altitude and this analysis indicates that a net-recombination and enhanced methane depletion is overcompensating the H2O destruction. Our finding is robust with good agreement between the two models."

New: "The photochemical depletion of $H_2O$ and shift to $H_2$ concentrations (e.g. Fig. 5.23, p. 312, Brasseur, 2005) clearly has no large effect at these emission altitudes. So instead to the expected removal of emitted $H_2O$ by photochemical depletion, we found a before unknown importance of the reaction rates of the net-recombination of $H_2O$ based on HOx recombination and an increased methane and nitric acid oxidation. Both models show an increase in $H_2O$ perturbation lifetime and $H_2O$ perturbation at the higher altitude, which is further increased by the net-recombination, i.e. overcompensation of photochemical depletion. Our finding is robust with good agreement between the two models."

**The debate on supersonic transport**

I realise that this is not a historical paper and I see that some mention has been made of earlier projects (e.g. COMESA). However, I suggest looking back a bit to the issue of supersonic transport (which is indeed discussed again today, see some of the citations in this paper). But in the seventies a controversy about supersonic transport had started in the United States. At that time, large fleets of stratospheric supersonic aircraft were planned (US: Boeing, Britain/France: Concorde, Soviet Union: Tupolev) and a fleet of 500 supersonic planes seemed a reasonable estimate. It is interesting to note that the concern was an enhanced catalytic ozone destruction; originally ozone destruction by OH and HO2 radicals (resulting from the release of water vapour in the engine exhausts, like discussed in this manuscript for hypersonic transport) was considered, but it was soon realised that the catalytic destruction of ozone by NOx posed a much greater threat to the ozone layer (Johnston, 1971; Crutzen, 1972). Indeed this issue was part of the motivation of Crutzen (1970) to investigate the impact of NOx on the ozone layer. Perhaps some effort to touch upon this history might be helpful to the paper.

We thank you very much for the additional information on the effect of NOx on the ozone layer. We included these for the readers next to the list of overview-publications in the introduction (line 43). We do not differentiate between emission like Kinnison et al or others. We rather look at the combined effect of $H_2O$, $H_2$ and NOx emission in this publication. Clearly, it is of interest to quantify the different sensitivities. However, the individual effects of emission are presented in a second publication, where impact of NOx emission, $H_2O$ emission and $H_2$ emission are looked at independently (Pletzer et al, in prep.).

**Some details**

- l. 3: it would be helpful to give approximate numbers for these emission.
  *Since we introduce only the concept of hypersonic aircraft, we would prefer to refrain from giving pieces of information here that depend on aircraft design and are based on multiple aircraft fleets with different ranges of emissions.*

- l. 6: if 15 km is in the tropics, months seems rater long, on the other hand months is short for emissions at (say) 30 km in the stratosphere. Perhaps one could be a bit more specific here.
  *Former: "While $H_2O$ that is emitted near the surface has a very short residence time (hours) and thereby no considerable climate impact, super- and hypersonic aviation emit at very high altitudes (15 km to 35 km), with residence times of months to several years, and therefore the emitted $H_2O$ has a substantial impact on climate via high altitude $H_2O$ changes."*
  *Now: "$H_2O$ that is emitted near the surface has a very short residence time (hours) and thereby no*

*considerable climate impact. Super- and hypersonic aviation emit at very high altitudes (15 km to 35 km) and water vapour residence times increase with altitude from months to several years with large latitudinal variations. Therefore, emitted $H_2O$ has a substantial impact on climate via high altitude $H_2O$ changes."*

- l. 8: I would not include (potential) speculations in the abstract –
  concentrate on the new findings of the paper.
  Thank you for sharing your impression. We changed the sentence accordingly.

- l. 350: what about the loss of H2O in the Antarctic stratosphere in
  winter (e.g., Kelly et al., 1989; Poshyvailo et al., 2018) could this loss
  process be of relevance for the considerations here? Is it implemented
  in the models?
  Since water vapour perturbations are low in the Antarctic, we do not expect a significant influence there. Polar stratospheric cloud chemistry is calculated in both models and contributes to not only dehydration, but denitrification as well. For EMAC, temperatures in the Antarctic polar stratospheric cloud environment are well represented. The simulated Antarctic polar vortex was relatively weak with a warm bias in former versions, but has been improved by changing the parameter that manages momentum deposition in the stratosphere and mesosphere by non-orographic gravity waves (Jöckel et al, 2016).

  We added the sentence to line 350: "Additionally, polar stratospheric clouds cause dehydration through sedimentation."

- l. 356: Here you say that higher altitudes have a negligible effect on
  the mass perturbation, but in l 361 you say that the "H2O mass per-
  turbation is approximately twice as large for the higher flying aircraft
  compared to the lower flying aircraft. . . " – isn't this a contradiction?
  I think this could be better explained.
  *Thank you for pointing that out. We slightly altered the sentences (changes underlined) in the first (line 356) and second paragraph (line 361) to avoid a mix-up with the previous paragraph.*

  *L 356: "The latter altitude range contains only a small amount of the mass perturbation in the models, since the largest mass perturbation accumulates in the middle and lower stratosphere, where air density is larger"*

  *L 361: "Absolute values of the mass perturbation and the respective perturbation lifetime of water vapour are listed in Table 3. Values were calculated for the perturbation above the tropopause (WMO, 1957). The total H2O mass perturbation is approximately twice as large for the higher flying aircraft compared to the lower flying aircraft for each model due to the longer transport to the troposphere and due to the larger emission. The perturbation lifetime clearly increases with cruise altitude from 2.8-3.5 years to 4.2-4.6 years."*

- l. 370, Fig 6: here and elsewhere: H2O should not be in italics in the
  figures.
  *H2O was changed accordingly in Fig. 6, Fig. 3, Fig. 8, Fig A2.*

- l. 424, Table 4: ozone in percent; do you mean total ozone here?
  Total ozone change
  *We are referring to the relative total mass change of ozone in %. We changed the caption accordingly.*

*"Relative total ozone mass change for each model and each aircraft fleet."*

- l. 522, Fig. 12: The figure shows an enhancement factor, which is not
  explained in the caption. No unit is given. However, a little below
  (l. 532) a unit is given in the text, and the values are compared to
  Fig. 12: I find this hard to follow.
  *The enhancement factor is unitless, since it is a ratio of radiative forcings from supersonic to subsonic aircraft. Clearly, the term used in l. 532 is not correct, since it is the normalized near-surface temperature change. The enhancement factor is the 61 in the next sentence. We added the information, that the enhancement factor has no unit in line 510-511 and corrected the sentences in line 532.*

- l. 575: This is not the most recent edition of this book; see Brasseur
  and Solomon (2005).
  *We updated the reference, the Fig. reference and the page number.*

- l. 617: How is this reference available?
  *The reference is available via* ls@vki.ac.be *or* secretariat@vki.ac.be*. The bibtex data was updated accordingly.*

- l. 634/637: Journal missing?
  *Journal was listed in the bib file. It is present now in the pdf as well.*

**References**

Frank, F., Jöckel, P., Gromov, S., & Dameris, M.: Investigating the yield of $H_2O$ and $H_2$ from methane oxidation in the stratosphere, Atmospheric Chemistry and Physics, 18, 9955–9973, doi: 10.5194/acp-18-9955-2018, URL https://www.atmos-chem-phys.net/18/9955/2018/ (2018)

Winterstein, F. & Jöckel, P.: Methane chemistry in a nutshell – the new submodels CH4 (v1.0) and TRSYNC (v1.0) in MESSy (v2.54.0), Geoscientific Model Development, 14, 661–674, doi: 10.5194/gmd-14-661-2021, URL https://gmd.copernicus.org/articles/14/661/2021/ (2021)

McKenzie, R. L., Matthews, W. A., & Johnston, P. V. The relationship between erythemal UV and ozone, derived from spectral irradiance measurements. Geophysical Research Letters, 18(12), 2269-2272, https://agupubs.onlinelibrary.wiley.com/doi/abs/10.1029/91GL02786 (1991).

---

## Editor Decision (ED1)

Editor comments on revised version of egusphere-2022-285

Pletzer et al.: The Climate Impact of Hydrogen Powered Hypersonic Transport

P2, L49: I would suggest to write "given by Grewe et al. (2010)" instead of "given in 2010 by Grewe et al."

P2, L49-50: This sentence is not really helpful. Add the references directly after the given numbers instead of having an additional sentence stating where you get these numbers from. Or if it is better to use two sentences for clarity then add these numbers to the second sentence.

P2, L59: "Polar dehydration within polar stratospheric clouds" sounds quite weird. Write "Polar dehydration caused by the sedimentation of polar stratospheric cloud particles……."

P3, L73: The "2" should be in subscript.

P3, L76: Same here for NOx. The x should be in subscript. This should be adjusted throughout the manuscript.

P3, L88: Move "yet" one line up and put it behind "not", so that it reads "not yet been assessed".

P3, L88: Move "as well" behind "remains"

P3, L91: add "the" so that it reads "on the impact" and add "atmospheric composition" or "stratospheric composition".

P3, L92: Move "flying at 30 km" at the end of the sentence and add "altitude" so that it reads "flying at 30 km altitude".

P4, L95: Add "the" -> "They focus on the sensitivity"

P4, L96: of -> in ? (Not sure which is correct, please check)

P4, L99: Abbreviation RF has not introduced yet.

P4, L108: Abbreviation "LAPCAT" and "PREPHA-type" has not been introduced. Further, the latter should be written in upright font.

P4, L114: section 4 -> Sect. 4, section 5 -> Sect. 5, section 6 -> Sect. 6 (use the Copernicus style)

P6, L147: Abbreviation MIPAS not introduced.

P6, L148-149: mode -> model? Anyway this is obsolete and should be deleted. Instead of "setup" it should rather read "tool".

P6, L157: Abbreviation ECHAM not introduced.

P6, L159: Abbreviation ECMWF not introduced.

P7, L176: Add "were" so that it reads "chemistry calculations were operated".

P7, L180: Add "to" so that it reads "and to alter specific humidity".

P7, L184: Add the link to the MESSy webpage and/or a reference to the latest version of the model.

P9, L239: Write STS and NAT rather than type I and type II since you are not explaining the different types. Write also what STS and NAT stands for and add the respective compositions.

P9, L240: write "includes sedimentation of the PSC particles and combine with the next sentence and continue with "which affect……."

P8, L252: Introduce the abbreviation ORCHIDEE and use and an upright font for ORCHIDEE.

P9, 259: Add trace gases "long lifetimes of trace gases in the stratosphere".

P9, L260ff: There are a lot of abbreviation that have not been introduced: IPCC, CMIP6, SSP, RCP CMIP5, SSP3-7.0.

General comment: The model description is quite long. You could consider to shorten it.

P11, L284: PREHA in upright font.

P12, Figure 2: Increase figure size.

P13, L317: section 5 -> Sect. 5

P14, L329: Abbreviation UTLS has not been introduced.

P15, L336: move the reference of Cohen behind "software".

P16, L372: Write instead of just sedimentation "sedimentation of particles" or "sedimenting particles"

P16, L373: resolved -> considered?

P18, R1: Use "Eq. 1" instead of "R1" (Thus use the Copernicus style) and use upright font for the chemical reactions.

P19, L416: Two "2" in H2 should be in subscript.

P19, L418: Same here for the "x" in HOx

P20, L419: use here the chemical abbreviations since you already have introduced them.

P23, L463: Add "(RF)" after "radiative forcing".

P23, L465ff and P24, L490ff: units should be written in an upright font.

P25, L524: considerabe -> considerable

P27, L576: Sect. 7.3 title Emission should read Emissions.

P28, L583: add "the" -> than "the" EMAC

P28, L591: Put numbers in subscript.

P28, L595: PREHA in upright font.

P29, L631: Add here the altitude in parenthesis once again.

P29, L631ff: "x" should be in subscript.

P30, L636: in -> at (thus it should read "at lower stratospheric altitudes".

Appendix.: Consider combining figure A6-A9 to one figure.

---

## Author Response (AR2)

**Authors Response**

Dear Mrs Khosrawi,

we would like to thank you very much for the list of technical improvements. Below you find your list with improvements. If the font is green we applied the improvements as suggested. Please that this applies to nearly all suggestions with some exceptions mostly related to abbreviations. Please do not hesitate to contact us again, if we misunderstood the suggestions regarding the abbreviations. Our answers are given in *blue below your suggestion.*

Yours sincerely,

Johannes Pletzer

**Editor comments on revised version of egusphere-2022-285**

Pletzer et al.: The Climate Impact of Hydrogen Powered Hypersonic Transport

P2, L49: I would suggest to write "given by Grewe et al. (2010)" instead of "given in 2010 by Grewe et al."

P2, L49-50: This sentence is not really helpful. Add the references directly after the given numbers instead of having an additional sentence stating where you get these numbers from. Or if it is better to use two sentences for clarity then add these numbers to the second sentence.

P2, L59: "Polar dehydration within polar stratospheric clouds" sounds quite weird. Write "Polar dehydration caused by the sedimentation of polar stratospheric cloud particles……."

P3, L73: The "2" should be in subscript.

P3, L76: Same here for NOx. The x should be in subscript. This should be adjusted throughout the manuscript.

P3, L88: Move "yet" one line up and put it behind "not", so that it reads "not yet been assessed".

P3, L88: Move "as well" behind "remains"

P3, L91: add "the" so that it reads "on the impact" and add "atmospheric composition" or "stratospheric composition".

P3, L92: Move "flying at 30 km" at the end of the sentence and add "altitude" so that it reads "flying at 30 km altitude".

P4, L95: Add "the" -> "They focus on the sensitivity"

P4, L96: of -> in ? (Not sure which is correct, please check)

*Thank you for pointing that out. "for" is the correct*

*choice. Now: "Their estimate for a reduction"*

P4, L99: Abbreviation RF has not introduced yet.

*The term RF was introduced in line 46. We happily*

*add another introduction, if needed.*

P4, L108: Abbreviation "LAPCAT" and "PREPHA-type" has not been introduced. Further, the latter should be written in upright font.

> The term LAPCAT was introduced in line 92. We further added the long term for PREPHA, which is "Programme de REcherche et de technologie sur la Propulsion Hypersonique Avancée"

P4, L114: section 4 -> Sect. 4, section 5 -> Sect. 5, section 6 -> Sect. 6 (use the Copernicus style)

P6, L147: Abbreviation MIPAS not introduced.

P6, L148-149: mode -> model? Anyway this is obsolete and should be deleted. Instead of "setup" it should rather read "tool".

P6, L157: Abbreviation ECHAM not introduced.

> The abbreviation was introduced in line
>
> 110-111

P6, L159: Abbreviation ECMWF not introduced.

P7, L176: Add "were" so that it reads "chemistry calculations were operated".

P7, L180: Add "to" so that it reads "and to alter specific humidity".

P7, L184: Add the link to the MESSy webpage and/or a reference to the latest version of the model.

> Added hyperlink to MESSy homepage

P9, L239: Write STS and NAT rather than type I and type II since you are not explaining the different types. Write also what STS and NAT stands for and add the respective compositions.

P9, L240: write "includes sedimentation of the PSC particles and combine with the next sentence and continue with "which affect……."

P8, L252: Introduce the abbreviation ORCHIDEE and use and an upright font for ORCHIDEE.

> ORCHIDEE is now upright. The abbreviation was introduced in line 125.

P9, 259: Add trace gases "long lifetimes of trace gases in the stratosphere".

P9, L260ff: There are a lot of abbreviation that have not been introduced: IPCC, CMIP6, SSP, RCP CMIP5, SSP3-7.0.

General comment: The model description is quite long. You could consider to shorten it.

> Thank you for the proposition. We get your point and would like to keep the full
>
> description in the manuscript.

P11, L284: PREHA in upright font.

P12, Figure 2: Increase figure size.

P13, L317: section 5 -> Sect. 5

P14, L329: Abbreviation UTLS has not been introduced.

P15, L336: move the reference of Cohen behind "software".

P16, L372: Write instead of just sedimentation "sedimentation of particles" or "sedimenting particles"

P16, L373: resolved -> considered?

P18, R1: Use "Eq. 1" instead of "R1" (Thus use the Copernicus style) and use upright font for the chemical reactions.

P19, L416: Two "2" in H2 should be in subscript.

P19, L418: Same here for the "x" in HOx

P20, L419: use here the chemical abbreviations since you already have introduced them.

P23, L463: Add "(RF)" after "radiative forcing".

P23, L465ff and P24, L490ff: units should be written in an upright font.

P25, L524: considerabe -> considerable

P27, L576: Sect. 7.3 title Emission should read Emissions.

P28, L583: add "the" -> than "the" EMAC

P28, L591: Put numbers in subscript.

P28, L595: PREHA in upright font.

P29, L631: Add here the altitude in parenthesis once again.

P29, L631ff: "x" should be in subscript.

P30, L636: in -> at (thus it should read "at lower stratospheric altitudes".

Appendix.: Consider combining figure A6-A9 to one figure.

We combined Figures A6, A7 and A8, A9 as pairs, since all combined

were too large for one page.

**Authors Response**

Dear Reviewer,

we would like to thank you very much for your second review and your list with minor suggestions. Below you find your text with suggestions below "Referee's comments". We applied most suggestions and our answers are given in *blue below the comments.*

Yours sincerely,

Johannes Pletzer

**Referee's comments**

General comments

I see that the authors have invested substantial effort in improving
the papers taking into account the comments received from all three
referees.

*Thank you for your kind words.*

I still have a few wording clarification suggestions (see below).
Should these minor points be adressed I recommend publication of the
paper.

===============================================
P. 1, l. 13: "leading to an increase in H2O concentrations." --> increase compared to what?

I suggest two sentences here:

...methane and nitric acid depletion. These processes lead to an
increase in H2O concentrations compared to a case with no emissions
from hypersonic aircraft.

P. 1 l. 14: increase --> increase with altitude (correct?)

p.1: l 16: suggest: 8-22%

p.1: l 17: suggest: 78-92%

*We applied all of the few wording clarification suggestions. Thank you very much for the propositions*
* * *
The paper now contains the following text:

New: "The photochemical depletion of H2O and shift to H2 concentrations (e.g. Fig. 5.23, p. 312, Brasseur, 2005) clearly has no large effect at these emission altitudes. So instead to the expected removal of emitted H2O by photochemical depletion, we found a before unknown importance of the reaction rates of the net-recombination of H2O based on HOx recombination and an increased methane and nitric acid oxidation. Both models show an increase in H2O perturbation lifetime and H2O perturbation at the higher altitude, which is further increased by the net-recombination, i.e. overcompensation of photochemical depletion. Our finding is robust with good agreement between the two models."

First: do you want to be explicit about the mechanisms of H2O by photochemical depletion? You could add the main chmeical processes or provide a citation.

    We added the most important reactions and referenced them in the text.

Second, I am not sure what "net-recombination of H2O based on HOx recombination" means. I could imagine that what is meant net production of water vaour based on the (radical recombination) reaction HO2 + OH and an increased methane (CH4+OH) and nitric acid oxidation (HNO3+OH) ...

    We restructured the description of the process.

    Now: "The photochemical depletion of H2O and shift to H2 concentrations (e.g. Fig. 5.23, p. 312, Brasseur, 2005) is clearly not limiting the water vapour perturbation lifetime at these emission altitudes. So instead to the expected removal of emitted H2O by photochemical depletion, we found a before unknown importance of water vapour recombination for hypersonic emissions. Several reactions including the hydroxyl radical actually overcompensate the photochemical depletion of H2O perturbations. The overcompensation results in a net-recombination (recombination-depletion > 0), that is driven by HOx recombination (mainly Eq. 4), an increased methane (Eqs. 5 and 6) and nitric acid oxidation (Eqs. 7 and 8). Both models show an increase in H2O perturbation lifetime and H2O perturbation at the higher altitude, which is further increased by the net-recombination. Our finding is robust with good agreement between the two models."

as it stands the text is confusing and I think that adding the actual reactions that are most relevant here helps.
(I think the issue is actually better described and discussed in the abstract)
* * *
The following text was added to the paper:

"The middle atmospheric balance of water vapour is determined by methane oxidation, photochemical lifetimes of HOx compounds and tropical upward transport, which is limited by the coldpoint temperature (LeTexier et al, 1998; Brasseur, 2005; Frank et al, 2018). Polar dehydration by polar stratospheric clouds and the sedimentation of the particles contribute to the balance."

First, it is good that the current papers are cited (Frank et al., 2018; Winterstein & Jöckel, 2021). But I do not understand why/how tropical upward transport is limited by the coldpoint temperature. Isn't it the temperature in the lower stratosphere which is relevant here rather than the temperature at one particular point? Alternatively, do you mean that stratospheric water vapour is influenced strongly by the entry value of water vapour? This enty value is indeed influenced by the cold point temperature. But then another wording/explanation is required (see also the cited papers).

> Thank you for pointing that out. We changed our choice of words accordingly (underlined).
>
> Now: "The middle atmospheric balance of water vapour is determined by methane oxidation, photochemical lifetimes of HOx compounds and _upward transport through the tropical upper troposphere lower stratosphere, which is limited by the cold temperatures_ (LeTexier et al, 1998; Brasseur, 2005; Frank et al, 2018). Polar dehydration by polar stratospheric clouds and the sedimentation of the particles contribute to the balance."

References

Brasseur and Solomon, 2005.
-- you cite two editions of this book in the paper
but I think you nedd only one citation

> Since the last revision our draft should only contain the 2005 version of Brasseur's book. If you would be so kind to explicitly state the line we will correct the citations gladly.

Frank, F., Jöckel, P., Gromov, S., & Dameris, M.: Investigating the yield of H2O and H2 from methane oxidation in the stratosphere, Atmospheric Chemistry and Physics, 18, 9955–9973, doi: 10.5194/acp-18-9955-2018, URL https://www.atmos-chem-phys.net/18/9955/2018/ (2018)

Winterstein, F. & Jöckel, P.: Methane chemistry in a nutshell – the new submodels CH4 (v1.0) and TRSYNC (v1.0) in MESSy (v2.54.0), Geoscientific Model Development, 14, 661–674, doi: 10.5194/gmd-14-661-2021, URL https://gmd.copernicus.org/articles/14/661/2021/ (2021)

---

## Author Response (AR3)

**Authors Response**

Dear Mrs Khosrawi,

Thank you very much for the list with technical corrections. Below you find your message, where we included our answers in *blue and italic*.

We also would like to thank you very much for guiding everyone through the submission process.

Yours sincerely,

Johannes Pletzer

**Comments to the author**:
Dear authors,

thanks for the time and effort you have put in the second revision of the manuscript. Before uploading your final files for publication I would like to ask you to consider the following technical corrections:

*We gladly consider the technical corrections and add the updates to our manuscript. Thank you for taking the time to increase the manuscripts quality.*

- L359: References to the figures are not appearing here correct

   *The references do appear correctly now.*

- L476: The Copernicus style is that multiplications are given by a dot rather than a star

   *We corrected the star with a \cdot. Now, the multiplications appear the right way.*

- L642: The "4" in CH4 and the "3" in HNO3 should be in subscript

   *We corrected the occurrences. Additionally, we added some missing subscripts in line 238-239.*

- Check the quality of Figure 4. On my screen the bottom and left plot frame did not appear correctly.

   *According to the email correspondence, we checked the Figures again. Lines in Figure 2 are visible without zooming.*

Best wishes, Farahnaz Khosrawi